# Spatial modeling reveals nuclear phosphorylation and subcellular shuttling of YAP upon drug-induced liver injury

Lilija Wehling[1,2], Liam Keegan[2], Paula Fernández-Palanca[1,3,4], Reham Hassan[5,6], Ahmed Ghallab[5,6], Jennifer Schmitt[1], Yingyue Tang[1], Maxime Le Marois[1], Stephanie Roessler[1], Peter Schirmacher[1], Ursula Kummer[2], Jan G Hengstler[5], Sven Sahle[2†], Kai Breuhahn[1*†]

[1]Institute of Pathology, University Hospital Heidelberg, Heidelberg, Germany; [2]Department of Modeling of Biological Processes, COS Heidelberg/BioQuant, Heidelberg University, Heidelberg, Germany; [3]Institute of Biomedicine (IBIOMED), University of León, León, Spain; [4]Centro de Investigación Biomédica en Red de Enfermedades Hepáticas y Digestivas (CIBERehd), Instituto de Salud Carlos III, Madrid, Spain; [5]Leibniz Research Centre for Working Environment and Human Factors, Department of Toxicology, Technical University Dortmund, Dortmund, Germany; [6]Department of Forensic Medicine and Toxicology, Faculty of Veterinary Medicine, South Valley University, Qena, Egypt

*For correspondence:
Kai.Breuhahn@med.uni-heidelberg.de

†These authors contributed equally to this work

**Abstract** The Hippo signaling pathway controls cell proliferation and tissue regeneration via its transcriptional effectors yes-associated protein (YAP) and transcriptional coactivator with PDZ-binding motif (TAZ). The canonical pathway topology is characterized by sequential phosphorylation of kinases in the cytoplasm that defines the subcellular localization of YAP and TAZ. However, the molecular mechanisms controlling the nuclear/cytoplasmic shuttling dynamics of both factors under physiological and tissue-damaging conditions are poorly understood. By implementing experimental in vitro data, partial differential equation modeling, as well as automated image analysis, we demonstrate that nuclear phosphorylation contributes to differences between YAP and TAZ localization in the nucleus and cytoplasm. Treatment of hepatocyte-derived cells with hepatotoxic acetaminophen (APAP) induces a biphasic protein phosphorylation eventually leading to nuclear protein enrichment of YAP but not TAZ. APAP-dependent regulation of nuclear/cytoplasmic YAP shuttling is not an unspecific cellular response but relies on the sequential induction of reactive oxygen species (ROS), RAC-alpha serine/threonine-protein kinase (AKT, synonym: protein kinase B), as well as elevated nuclear interaction between YAP and AKT. Mouse experiments confirm this sequence of events illustrated by the expression of ROS-, AKT-, and YAP-specific gene signatures upon APAP administration. In summary, our data illustrate the importance of nuclear processes in the regulation of Hippo pathway activity. YAP and TAZ exhibit different shuttling dynamics, which explains distinct cellular responses of both factors under physiological and tissue-damaging conditions.

## Editor's evaluation

The Hippo signaling pathway is essential for multiple physiological processes, most notably the regulation of cell proliferation and survival during wound healing. Wehling et al. provide a molecular framework for an alternative mechanism by which the Hippo effector molecule YAP's sub-cellular localization is regulated by cell compartment-specific phosphorylation. Specifically, the authors demonstrate dynamic regulation of shuttling of YAP both in vitro and in vivo during drug-induced

liver injury. Given the importance and developmental conserveness of the Hippo pathway, the work is of broad interest to the field of developmental and regenerative biology.

## Introduction

The evolutionary conserved Hippo signaling pathway controls tissue homeostasis through the regulation of cell proliferation, apoptosis, differentiation, and cellular fate (*Zhao et al., 2011*). Activity of this pathway is affected by information derived from extracellular matrix stiffness, actomyosin dynamics, and cell density (*Deng et al., 2015*; *Zhao et al., 2007*), which in turn modulate a serine/threonine kinase cassette consisting of serine/threonine kinase 3/4 (STK3/4; MST1/2) and large tumor suppressor 1/2 (LATS1/2). According to the canonical Hippo pathway model, active cytoplasmic LATS1/2 phosphorylate and inactivate two important downstream pathway effectors: the transcriptional co-activators yes-associated protein (YAP) and its paralog transcriptional coactivator with PDZ-binding motif (TAZ). This phosphorylation is associated with nuclear YAP/TAZ exclusion followed by their proteasomal degradation. In contrast, Hippo pathway inactivation in the cytoplasmic compartment causes YAP/TAZ dephosphorylation and nuclear translocation. In the nucleus, YAP/TAZ bind DNA sequence-specific transcription factors such as TEA DNA-binding proteins (TEADs) or forkhead box protein M1 (FOXM1), which control the transcription of genes involved in for example, cell cycle control and paracrine communication (*Marti et al., 2015*; *Thomann et al., 2020*; *Weiler et al., 2017*).

Due to its pivotal role in the regulation of cell proliferation, the Hippo/YAP/TAZ axis is important for tissues maintenance during regenerative processes. Indeed, YAP- or YAP/TAZ deficiency reduce the regenerative capacity of tissues such as liver, skin, and heart (*Lee et al., 2014*; *Lu et al., 2018*; *Xin et al., 2013*). As exemplified in detail for the liver, YAP is activated in hepatocytes under disease conditions that support a continuous regenerative response in different in vivo model systems (*Machado et al., 2015*; *Mooring et al., 2020*). Interestingly, the roles of YAP and TAZ are not identical, since TAZ, but not YAP, contributes to fat accumulation in the liver (steatosis) (*Wang et al., 2016*). In contrast, YAP protects from liver ischemia–reperfusion injury by promoting tissue repair (*Liu et al., 2019*). These findings strongly suggest a cell-protective role of the Hippo pathway under liver injury conditions; however, the function of YAP and TAZ in this biological process may differ. As YAP and TAZ control the expression of similar target genes in different cell types (*Weiler et al., 2020*; *Zanconato et al., 2015*), other regulatory mechanisms must account for these phenotypic differences such as differential nuclear/cytoplasmic shuttling (*Reggiani et al., 2021*). However, detailed comparative studies regarding subcellular localization dynamics for YAP and TAZ are missing as well as spatially resolved computational framework such as partial differential equation (PDE) models.

Recent data demonstrated a direct impact on YAP by acetaminophen (APAP), which is a widely used analgesic drug and leading cause of acute liver failure (*Poudel et al., 2021*). Here, APAP overdose in mice caused liver injury, which was associated with a prominent nuclear YAP enrichment in hepatocytes. Interestingly, silencing of YAP did not impair liver regeneration (which would be expected after inactivation of this proproliferative factor), but promoted tissue regeneration upon APAP-induced liver damage (*Poudel et al., 2021*). In contrast, a supportive role of YAP in tissue repair and regeneration was suggested by chemical blockade of upstream Hippo pathway kinases MST1/2, which reverted APAP-induced liver damage (*Fan et al., 2016*). These results point to intricate and context-specific roles of YAP and/or TAZ under regenerative conditions.

Our study comprises a quantitative and mechanistic analysis of YAP/TAZ dynamics based on spatially resolved, high-throughput confocal live cell imaging data and PDE modeling. By using this experimental and computational toolbox, we show that nuclear phosphorylation of YAP/TAZ is important for their subcellular shuttling dynamics and that both factors do not equally respond to cell density and APAP-induced hepatocellular damage. We mechanistically show that APAP overdose stimulates a rapid reactive oxygen species (ROS) response, which facilitates the physical nuclear interaction of YAP and AKT as predicted by computational modeling. Thus, our data demonstrate for the first time that APAP-induced YAP activation is a dynamic and specific process that relies on the sequential activation of molecular events.

## Results

### Establishment of an in vitro model for measuring time-resolved spatial localization of YAP and TAZ in hepatocellular cells

Due to technical limitations, primary human hepatocytes are not suitable for the analysis of dynamic YAP/TAZ shutting in vitro: they rapidly undergo trans-differentiation in culture and different human donors exhibit genomic/epigenetic variability, which diminish reproducibility. We therefore selected the hepatocyte-derived liver cancer cell line Hep3B that was characterized by a prominent YAP/TAZ and cytochrome P450 2E1 (CYP2E1) expression (*Figure 1—figure supplement 1A*; *Weiler et al., 2020*). Detectable CYP2E1 levels are required for the enzymatic transformation of APAP to toxic *N*-acetyl-*p*-benzoquinone imine (NAPQI) (*Lee et al., 1996*).

Since we were interested in cell-to-cell variability and spatial localization information of YAP and TAZ, we established a cell line that allowed the quantitative and spatiotemporal investigation of subcellular YAP/TAZ distribution near the single-cell resolution. For this, we generated Hep3B cells that stably express mVenus-tagged YAP (mVenus-YAP) and mCherry-tagged TAZ (mCherry-TAZ) (*Figure 1A*). In addition, mCerulean-tagged histone H2B (mCerulean-H2B) allowed the spatial allocation of YAP/TAZ. These chosen fluorophores facilitated optimal discrimination between emission spectra by live cell microscopy (*Figure 1—figure supplement 1B*). Subcellular localization changes of tagged YAP and TAZ under variable cell density conditions illustrated functionality of both proteins regarding nuclear–cytoplasmic shuttling (*Figure 1B*). However, we observed a high degree of intracellular variability.

For analysis of variable high-throughput data, we developed an algorithm for the semiquantitative measurement of tagged YAP and TAZ in nuclei and cytoplasm. In brief, spatial image data were classified based on the expression of mVenus-YAP and mCherry-TAZ using the Weka segmentation algorithm (*Figure 1C*, *Figure 1—figure supplement 1C*; *Arganda-Carreras et al., 2017*). The fluorophore intensity measurements were used to calculate the nuclear/cytoplasmic ratio (NCR), which characterized the relative nuclear enrichment of YAP or TAZ. Therefore, this approach allowed us to simultaneously define slightest subcellular changes of YAP and TAZ in small cell populations (field of view, FOV) as well as individual cells under live cell conditions.

Subsequently, the fluorophore-expressing cells were grown under variable cell density conditions and the YAP/TAZ localization was measured by confocal microscopy. The results confirmed that tagged YAP and TAZ dynamically shuttled between cytoplasm and the nucleus depending on abundance of cell–cell contacts (cell density) (*Figure 1D*). However, TAZ showed a considerably less pronounced dynamic behavior compared to YAP (*Figure 1D*). YAP was predominantly detectable in cell nuclei under low cell density conditions (NCR > 1), while increasing cell density led to cytoplasmic enrichment of both factors (NCR < 1). Using this algorithm, cell density-dependent protein shuttling was also demonstrated for other liver cancer cells (*Tóth et al., 2021*).

Together, the confocal imaging and image processing pipeline efficiently detects subtle differences in the dynamic subcellular changes of YAP and TAZ.

### Mathematical modeling predicts nuclear phosphorylation of YAP and TAZ

To investigate why Hippo pathway effectors YAP and TAZ differently respond to a low cell density and which Hippo pathway topology can explain the observed localization differences, we mathematically modeled the Hippo pathway using a PDE modeling framework (for detailed description of the PDE modeling approach refer to Materials and methods and Appendix 1).

First, we investigated a canonical Hippo pathway model, where phosphorylation takes place in the cytoplasm and where unphosphorylated YAP/TAZ shuttle to the nucleus (*Figure 2A*). However, this mathematical model was not able to reproduce the localization patterns which were observed in the experimental data (*Figure 2B*). For example, the model cannot sufficiently explain the distribution of YAP and TAZ in cell nuclei or at the nuclear membrane. In detail, the cell area around the nuclear envelope on the cytoplasmic side strongly underrepresented the concentration of YAP/TAZ, as indicated by the blue pixels in the residual image (experimental data minus model simulation). Moreover, the simulated YAP/TAZ distribution pattern of the canonical model showed a decreased protein concentration in the center of the nucleus, with increasing gradient toward the nuclear envelope. This indicates that the experimental observation is dominated by a protein that is disseminated from the

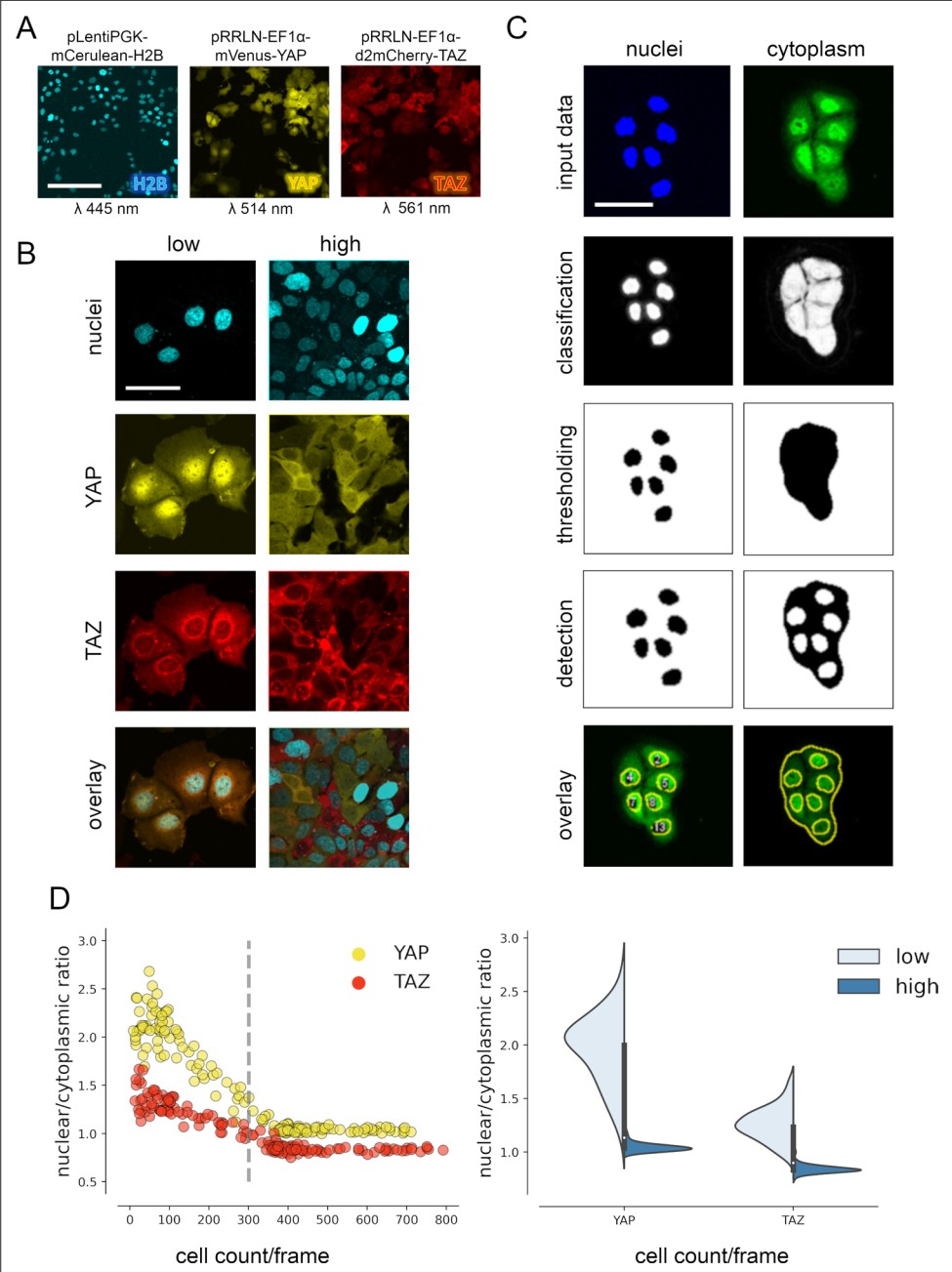

**Figure 1.** YAP and TAZ show distinct nuclear shuttling in hepatocellular cells upon increasing cell density. (**A**) The hepatocyte-derived cell line Hep3B was transduced using three lentiviral vectors coding for mCerulean-tagged H2B (pLentiPGK-mCerulean-H2B), mVenus-tagged YAP (pRRLN-EF1α-mVenus-YAP), and mCherry-tagged TAZ (pRRLN-EF1α-d2mCherry-TAZ). Combined treatment with the antibiotics hygromycin, geneticin, and blasticidin selected for triple-positive cells. Cells were confocally imaged in three channels: 445 nm (CFP, mCerulean), 514 nm (YFP, mVenus), and 561 nm (RFP, mCherry). Scale bar: 250 μm. (**B**) Exemplary pictures illustrating the subcellular localization of H2B, YAP, and TAZ proteins under low (left) and high (right) cell density conditions. Scale bar: 50 μm. (**C**) Automatic image analysis workflow depicts analysis of nuclear (left) and cytoplasmic (right) fluorescence intensity in confocal images of living cells. The acquired images were prescreened for imaging artifacts (e.g., out-of-focus images were excluded) and subjected to the image analysis pipeline in Fiji. Object classification (Weka algorithm), thresholding, and object counting were performed. Object masks were overlaid with original data and the nuclear/cytoplasmic ratio (NCR) was calculated by dividing average fluorescent signal intensity of YAP/TAZ in the nucleus with the fluorescence intensity in the cytoplasm. Scale bar: 50 μm. (**D**) Quantification of YAP and TAZ NCR under increasing cell density conditions. One of four representative experiment is depicted. Left: the NCR of

*Figure 1 continued on next page*

*Figure 1 continued*

YAP (yellow, *n* = 310) and TAZ (red, *n* = 310) was plotted against cell count per visual field (0.4 mm²). A high NCR value indicates predominant nuclear localization. Dashed line shows mean cell count. Right: violin plots summarize shift in NCR between low (below mean cell count) and high (above mean cell count) cell density for YAP and TAZ.

The online version of this article includes the following source data and figure supplement(s) for figure 1:

**Figure supplement 1.** In vitro model establishment.

**Figure supplement 1—source data 1.** Original western blot data.

nucleus and undergoes diffusion and degradation. The canonical Hippo pathway cannot explain this effect, illustrated by residuals (*Figure 2B*).

This let us conclude that alternative model topologies must be considered to explain the experimental data. We therefore generated an alternative PDE model, which precisely described YAP/TAZ localization as observed in the image data. In this model, a phosphorylation and dephosphorylation reaction of YAP and TAZ in the nucleus was included (*Figure 2C,D*, *Figure 2—figure supplement 1A, B*). Moreover, the alternative model describes that unphosphorylated YAP/TAZ are transported to the nucleus and pYAP/pTAZ are rapidly excluded from the nucleus, which agrees with the general consensus that phosphorylated YAP and TAZ predominantly localize in the cytoplasm (*Figure 2—figure supplement 1C*). Using well-fitting parametrization of the model, the simulated concentration pattern was dominated by phosphorylated YAP/TAZ disseminating from the nucleus, which now matched the observed experimental data well (*Figure 2D*).

Since YAP and TAZ are considered biochemically similar molecules, the question arises how the pronounced observed differences in the spatial distribution patterns can be explained. Using the alternative Hippo pathway model, we demonstrated that the different phenotypes could be reproduced with one single PDE model where only the parameters of the YAP/TAZ phosphorylation and dephosphorylation reactions in the nucleus differed, while all other parameters (including nuclear import and export reactions) were kept the same (*Figure 2E*, *Figure 2—figure supplement 1D*).

In detail, low nuclear phosphorylation/dephosphorylation ratio (0.17) induced nuclear localization of YAP, while high ratio (0.75) induced nuclear protein exclusion. Upon increased phosphorylation activity (>0.2), the nuclear fraction of TAZ decreased and the summarized residual term for TAZ was reduced. According to residual terms, best accordance between the model and the experimental data was achieved at low phosphorylation/dephosphorylation ratio for YAP and at higher phosphorylation/dephosphorylation ratio for TAZ (*Figure 2E*, right panel). These findings suggest that differences in the nuclear phosphorylation rates of YAP and TAZ can explain the distinct shuttling patterns of both factors (as observed in *Figure 1B*). In addition, TAZ may require stronger phosphorylation than YAP to induce its efficient shuttling.

The results of the alternative mathematical model required that major components for YAP/TAZ phosphorylation process were present in the cell nucleus. Indeed, subcellular fractionation experiments illustrated that the Hippo pathway kinases LATS1/2 (total and phosphorylated) were predominantly localized in nuclear protein fraction (*Figure 2F*). Moreover, we and others reported that nuclear LATS1/2 can control YAP phosphorylation and localization (data not shown) (*Ege et al., 2018*; *Li et al., 2014*). The relevance of this nuclear interaction is substantiated by a proximity ligation assay (PLA), which illustrated that pLATS1/2 physically interacted with YAP not only in the cytoplasm but also in the nucleus (*Figure 2G*). To quantitatively compare the number of YAP/pLATS interactions in cell nuclei and cytoplasm, we utilized a computational algorithm counting signals in nuclear and cytoplasmic area of the cells (*Figure 2—figure supplement 2A, B*). The signal quantification confirmed that YAP and pLATS interactions were detectable in both compartments with significantly higher interaction count in cell nuclei (*Figure 2H*).

In summary, our findings suggest that nuclear YAP/TAZ phosphorylation contributes to their inactivation and nuclear exclusion. Moreover, variable phosphorylation rates of YAP and TAZ explain the subcellular shuttling differences between both factors.

## APAP regulates YAP protein localization and activity

To test if our findings are of relevance under conditions where YAP and TAZ are actively regulated, we decided to use a drug-induced liver injury (DILI) model. For this, we treated Hep3B cells expressing

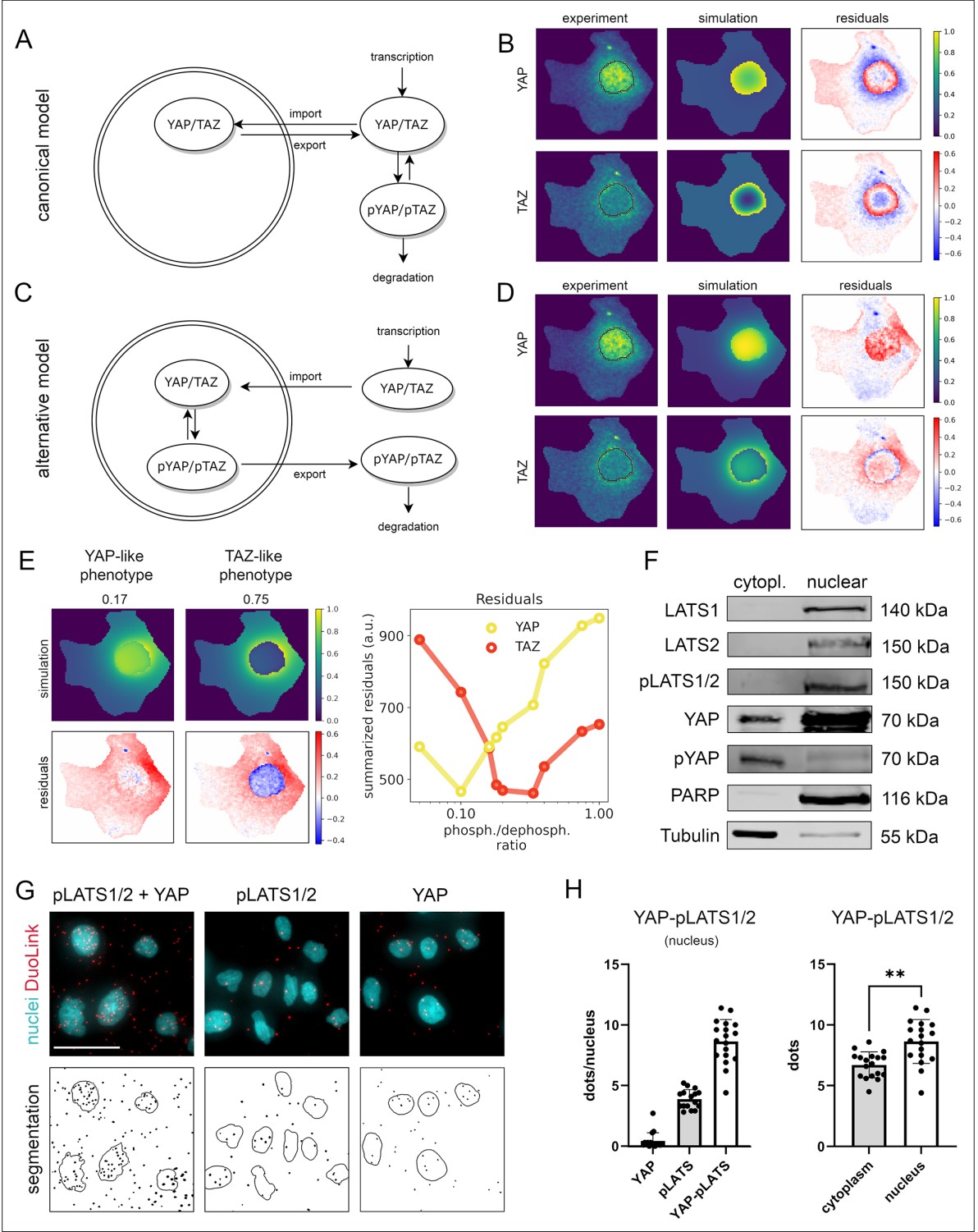

**Figure 2.** Mathematical modeling predicts that nuclear phosphorylation controls YAP/TAZ subcellular localization. (**A**) The model reaction scheme of the canonical Hippo pathway. Only unphosphorylated YAP/TAZ can enter the nucleus, whereas pYAP and pTAZ are exclusively localized to the cytosol. The phosphorylation of YAP/TAZ takes place exclusively outside nuclei. (**B**) Partial differential equation (PDE) model simulation of the canonical Hippo pathway compared to the experimentally measured YAP/TAZ localization. The residuals (experimental data minus model simulation) indicate low spatial accordance of the model simulation with the experimentally measured subcellular localization of YAP/TAZ. Images were normalized to the maximal value within each image. Nuclear outline is indicated in black. (**C**) The model reaction scheme of the alternative Hippo pathway model. Phosphorylation and dephosphorylation of YAP/TAZ take place in the nucleus. Unphosphorylated YAP/TAZ is imported in the nucleus and phosphorylated YAP/TAZ is

*Figure 2 continued on next page*

*Figure 2 continued*

rapidly exported to the cytoplasm. (**D**) PDE simulation of the alternative model compared to experimental data. Residuals for the subcellular localization (e.g., nuclear distribution) sufficiently reflect results from confocal microscopy. Nuclear outline is indicated in black. (**E**) Simulated impact of the nuclear phosphorylation to dephosphorylation ratio on subcellular localization of YAP in the alternative Hippo pathway model. Left: model simulation of two phosphorylation/dephosphorylation rates (0.17 and 0.75). Right: summarized residuals with respect to experimental data of YAP and TAZ as a function of phosphorylation to dephosphorylation ratio in the nucleus. (**F**) Western immunoblot after nuclear and cytoplasmic protein fractionation under low cell density conditions. The central Hippo pathway kinases LATS1/2 were detectable in the nuclear fraction. As expected, high pYAP levels are detectable in the cytoplasm, illustrating that the protein is transported outside the nucleus upon phosphorylation. PARP and Tubulin serve as loading controls for nuclear and cytoplasmic fractions, respectively. Equal amounts of cytoplasmic and nuclear proteins were loaded (*n* = 4; one representative experiment shown). (**G**) Proximity ligation assay (PLA) for phosphorylated LATS1/2 (pLATS1/2) and YAP (top row). Red dots indicate physical interaction between pLATS1/2 and YAP. DAPI (4′,6-diamidino-2-phenylindole)-stained nuclei are indicated in cyan. Individual pLATS and YAP antibodies serve as assay controls. Bottom row: computational segmentation of nuclei (empty circles) and dots (black spots). Scale bar: 50 µm. (**H**) Quantification of the PLA assay. Each dot represents an individual image. Left: quantification of interactions between YAP and pLATS1/2 in the nucleus (*n* = 18) compared to the negative controls (pLATS1/2 and YAP antibodies alone, *n* = 16 and *n* = 17, respectively). Right: the number of PLA dots of pLATS1/2 and YAP interaction in cytoplasm and in nuclei (*n* = 18). Statistical test: two-tailed paired *t*-test (p value = 0.01) **p ≤ 0.01.

The online version of this article includes the following source data and figure supplement(s) for figure 2:

**Source data 1.** Raw data for Western Blot shown in *Figure 2F*.

**Figure supplement 1.** Parametrizing alternative computational steady-state models.

**Figure supplement 2.** Data analysis pipeline for proximity ligation assay (PLA) quantification.

---

tagged mVenus-YAP and mCherry-TAZ with APAP (10 nM) (*Barbier-Torres et al., 2017*), and quantitatively investigated the dynamic shuttling of both factors.

Image quantification revealed that APAP led to a gradual and time-dependent nuclear enrichment of YAP after 48 hr, while TAZ only weakly responded to APAP (*Figure 3A,B*). This APAP-induced nuclear shuttling effect was clearly detectable under high cell density culture conditions, which is characterized by nuclear YAP/TAZ exclusion. Effects on YAP were less pronounced 24 hr after APAP administration (*Figure 3—figure supplement 1A,B*). No obvious response was detectable for YAP and TAZ at earlier time points post APAP treatment (data not shown). Severe effects of APAP on cell toxicity and apoptosis in the chosen experimental setup were excluded by measuring cell viability and PARP cleavage (data not shown).

The APAP-induced nuclear enrichment of YAP was confirmed with cell population-based Western immunoblotting followed by detection of total and phosphorylated YAP (*Figure 3C*, for quantification of all western blots see *Figure 3—figure supplement 2*). Notably, at early time points YAP hyperphosphorylation indicated protein inactivation (up to 3 hr after APAP administration). However, between 24 and 48 hr after APAP treatment, a clear YAP dephosphorylation was detectable (*Figure 3C*). Similar results were observed for another hepatocyte-derived cell line (*Figure 3—figure supplement 1C*).

To test if APAP-dependent dephosphorylation/activation of YAP at later time points is leading to its transcriptional activation, we measured the relative expression of known YAP target genes cysteine-rich angiogenic inducer 61 (*CYR61*) and ankyrin repeat domain 1 (*ANKRD1*) by real-time PCR. Indeed, we observed induction of these genes already 24 hr after APAP treatment, which reflected the earliest activation of YAP (*Figure 3D*). To confirm YAP dependence on the target gene induction and to exclude YAP-independent effects on gene expression, we performed additional rescue experiments. For this, the expression of YAP was silenced by RNA interference (RNAi) in APAP-treated cells followed by the measurement of YAP target genes. According to our hypothesis, YAP inhibition partly abolished the APAP-dependent induction of *CYR61* and *ANKRD1* (*Figure 3E*).

Together, these data demonstrate that APAP predominantly acts on the Hippo pathway effector YAP in a bimodal manner. Upon APAP treatment an immediate phosphorylation/inactivation of YAP is followed by its late dephosphorylation/activation.

## APAP controls YAP phosphorylation via ROS and AKT

Based on previous data, we hypothesized a mechanistic connection between APAP-induced ROS (*Barbier-Torres et al., 2017*; *Shuhendler et al., 2014*) and AKT-driven YAP phosphorylation (*Basu et al., 2003*; *Romano et al., 2010*). This mechanistic link was investigated at early time points after APAP treatment (up to 3 hr) to exclude unspecific effects caused by APAP at later time points (e.g., due to secondary and/or unspecific APAP effects).

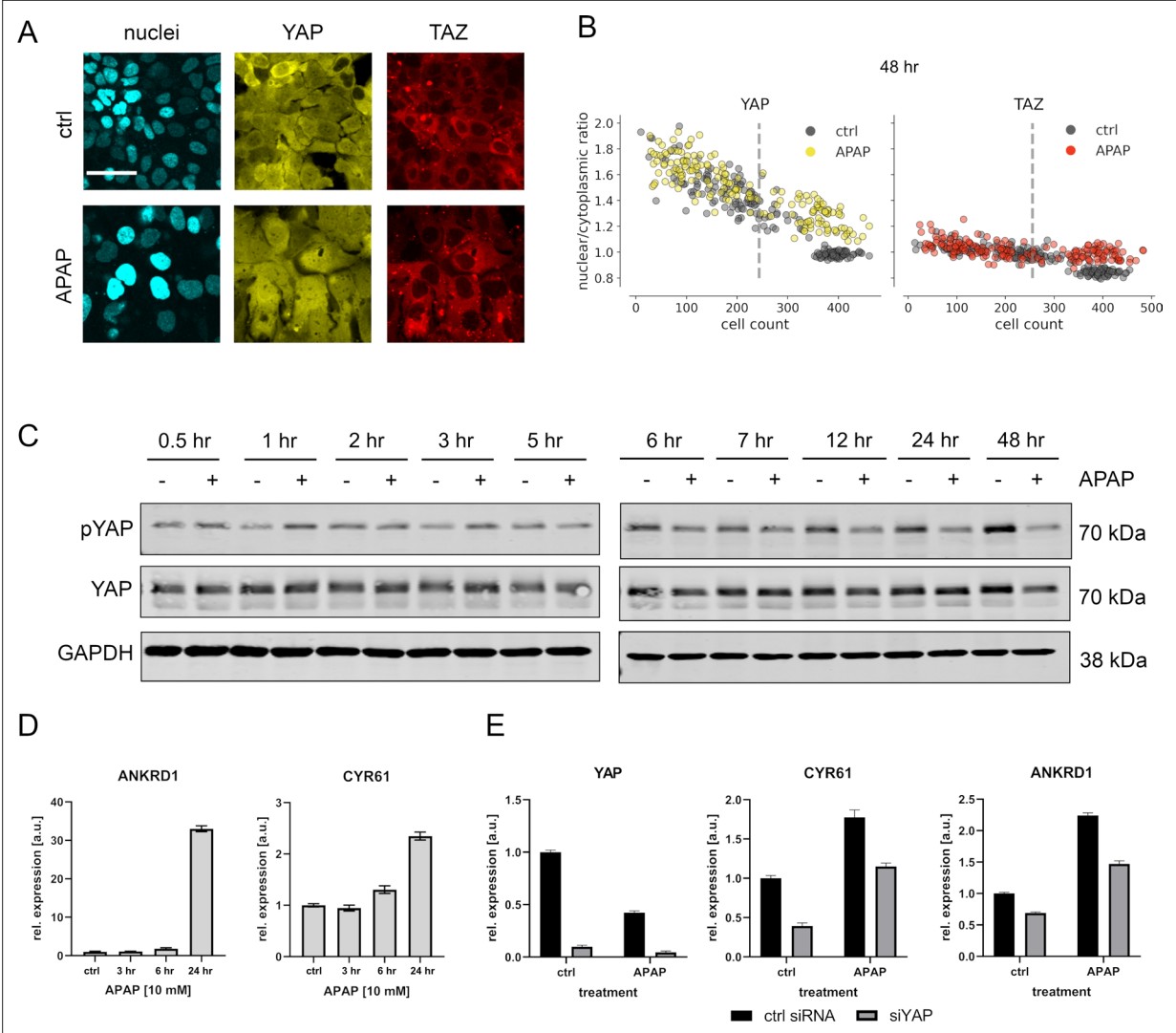

**Figure 3.** APAP regulates YAP phosphorylation, localization, and expression of its target genes. (**A**) Live cell confocal imaging of H2B-mCerulean, YAP-mVenus, and TAZ-mCherry in Hep3B cells. Upper row: control treatment (phosphate-buffered saline, PBS). Lower row: APAP treatment (10 mM) induces nuclear enrichment of YAP, but not TAZ protein after APAP treatment within 48 hr (*n* = 4; one representative experiment shown). Scale bar: 50 μm. (**B**) Live cell confocal microscopy image quantification. Nuclear/cytoplasmic ratio (NCR) of YAP and TAZ with APAP (10 mM for 48 hr; yellow and red circles, *n* = 180 per channel) and control (PBS; gray circles, *n* = 180 per channel) treatment under increasing cell density conditions (represented as mean cell count per visual field). Each dot represents a single visual field. Dashed line: mean cell density. (**C**) Western immunoblot of YAP and pYAP after APAP treatment (10 mM) in Hep3B cells (*n* = 6; one representative experiment shown). GAPDH served as a loading control. (**D**) Relative expression of the YAP target genes *ANKRD1* and *CYR61* 24 hr after APAP (10 mM) treatment in HLF cells (*n* = 2; one out of two biological replicates shown). (**E**) Rescue experiment: relative expression of YAP and its target genes *CYR61* and *ANKRD1* with and without siRNA-mediated YAP inhibition (for 24 hr) with or without APAP (10 mM) treatment for 24 hr (*n* = 2; one out of two biological replicates shown).

The online version of this article includes the following source data and figure supplement(s) for figure 3:

**Source data 1.** Source data for Western Blot shown in *Figure 3C*.

**Figure supplement 1.** APAP treatment of hepatocellular cells induces YAP and TAZ dynamics.

**Figure supplement 1—source data 1.** Source data for Western Blot in *Figure 3—figure supplement 1C*.

**Figure supplement 2.** Quantification of the Western immunoblot analysis of *Figures 3 and 4*.

Western immunoblotting showed that both YAP and AKT were phosphorylated early after APAP incubation (*Figure 4A*), which was in accordance with our previous findings (*Figure 3C*). In parallel, an ROS detection assay revealed that APAP-induced prominent ROS activity comparable to the known ROS inducers hydrogen peroxide ($H_2O_2$) and tert-butyl hydroperoxide (TBHP) in hepatocellular

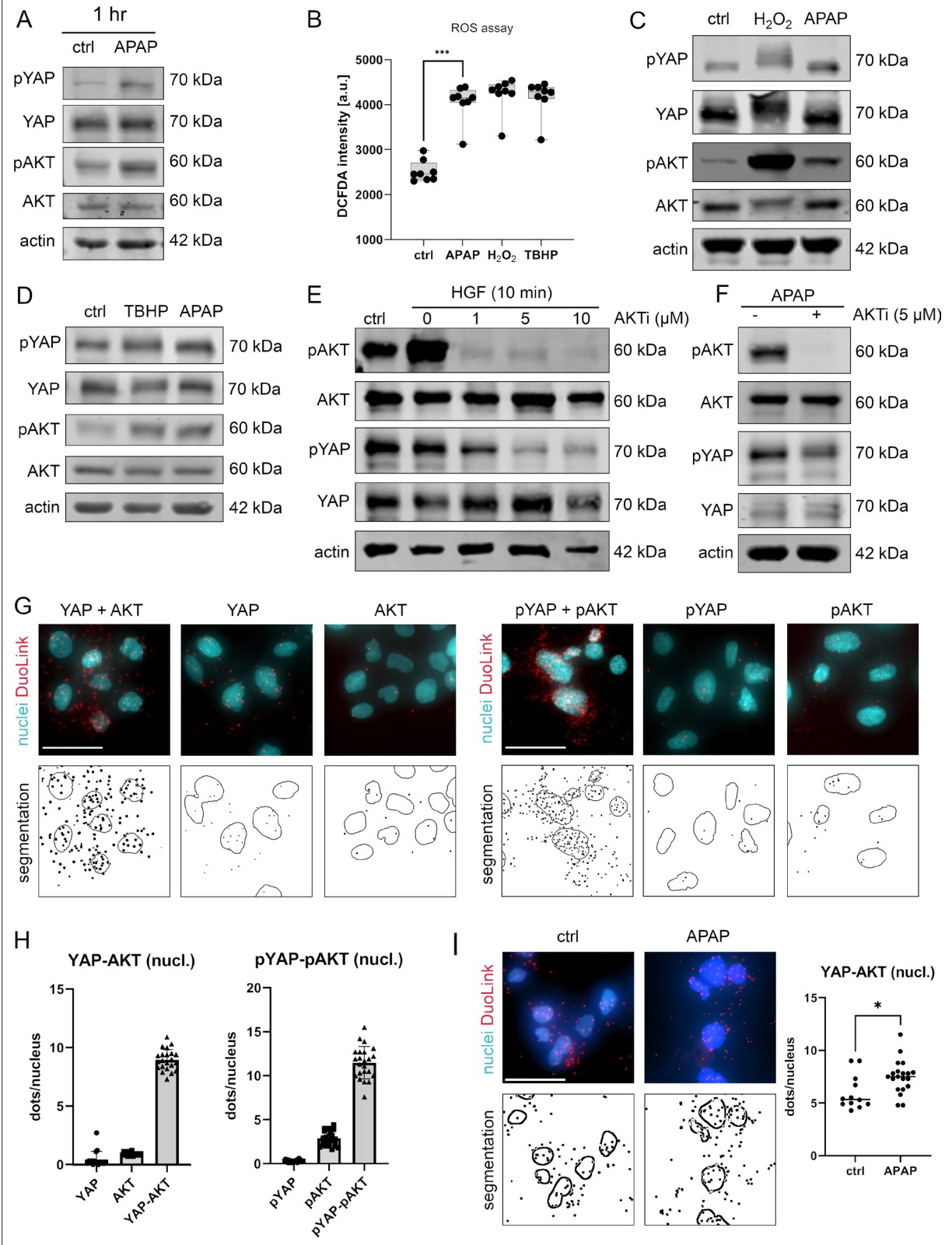

**Figure 4.** APAP controls nuclear YAP enrichment via induction of reactive oxygen species (ROS) and AKT. (**A**) Western immunoblot analysis of pYAP, YAP, pAKT, and AKT after APAP (10 mM) administration in Hep3B cells after 1 hr (Hep3B). The concentration of pYAP and pAKT is induced after APAP treatment in comparison to phosphate-buffered saline (PBS)-treated cells (ctrl); $n = 2$. (**B**) Spectrophotometric ROS measurement after APAP (10 mM), hydrogen peroxide ($H_2O_2$, 2 mM), and tert-butyl hydroperoxide (TBHP, 300 μM) treatment in Hep3B cells after 6 hr ($n = 8$ technical replicates, one out

*Figure 4 continued on next page*

*Figure 4 continued*

of two biological replicates is shown). APAP as well as $H_2O_2$ and TBHP induce ROS formation in living cells. Statistical test: one-way analysis of variance (ANOVA) with Geisser–Greenhouse correction (adjusted p value = 0.001) ***p-value ≤ 0.001. Whiskers depict min and max of the dataset. (C) Western immunoblot analysis of pYAP, YAP, pAKT, and AKT after $H_2O_2$ (2 mM) and APAP (10 mM) treatment for 1 hr. APAP, as well as $H_2O_2$ induce YAP and AKT phosphorylation compared to untreated cells (ctrl); n = 4. (D) Western immunoblot analysis of pYAP, YAP, pAKT, and AKT dynamics after TBHP (300 µM) and APAP (10 mM) treatment for 2 hr (Hep3B). APAP and TBHP induce AKT and YAP phosphorylation as compared to respective controls (ctrl). (E) Western immunoblot analysis of pAKT, AKT, pYAP, and YAP after hepatocyte growth factor (HGF) treatment (10 ng/µl) for 10 min (Hep3B). The cells were pretreated with an AKT inhibitor VIII (AKTi, 1, 5, and 10 µM) and starved for 3 hr prior HGF administration. Data illustrate that AKT inhibition prevents HGF-induced AKT and YAP phosphorylation. (F) Western immunoblot of pAKT, AKT, pYAP, and YAP after APAP (10 mM) administration for 1 hr (Hep3B). Cells were starved in Fetal calf serum (FCS)-free medium and pretreated with AKTi (5 µM) before APAP treatment for 3 hr. Results demonstrate that AKT inhibition reduces YAP phosphorylation early after APAP treatment. (G) Proximity ligation assay (PLA) of the protein combinations YAP/AKT and pYAP/pAKT. Red dots indicate interactions between YAP/AKT and pYAP/pAKT, respectively. DAPI-stained nuclei are depicted in cyan. Single YAP, AKT, pYAP, and pAKT stains serve as assay controls. Bottom row: segmentation of the nuclei (empty circles) and dots (black spots). One out of two representative experiment is shown. Scale bar: 50 µm. (H) Quantification of the interactions between YAP/AKT and pYAP/pAKT in the nucleus (experiment shown in G). One symbol represents one image (YAP = 17, AKT = 16, YAP-AKT = 22, pYAP = 24, pAKT = 25, pYAP-pAKT = 23). (I) PLA showing YAP–AKT interaction units per nuclei of untreated (ctrl, n = 12) and APAP-treated cells (10 mM, for 2 hr, n = 20). APAP treatment induces nuclear interaction between YAP and AKT. Statistical test: unpaired two-tailed parametric t-test (p value = 0.0155). For A and C–F, actin served as loading control.

The online version of this article includes the following source data for figure 4:

**Source data 1.** Source data for Western Blots shown in ***Figure 4A,C-F***.

cells (***Figure 4B***). Moreover, treatment of cells with ROS inducers demonstrated a clear (with $H_2O_2$) or moderate (with TBHP) induction of AKT and YAP phosphorylation after 1 and 2 hr, respectively (***Figure 4C,D***, quantification of western blots in ***Figure 3—figure supplement 2***). These data indicated that APAP regulated ROS activity, which itself controlled AKT and YAP activity.

To confirm the rapid and AKT-dependent phosphorylation of YAP, liver cells were treated with hepatocyte growth factor (HGF), which activates/phosphorylates AKT kinases via binding the receptor c-MET within 10 min (***Xiao et al., 2001***). HGF administration led to a clear phosphorylation of AKT but not YAP, which might be caused by saturation effects (no further YAP phosphorylation possible under given culture conditions) (***Figure 4E***). However, simultaneous inhibition of AKT kinase activity by the specific inhibitor AKTi (***Adlung et al., 2017***; ***Barnett et al., 2005***), not only abolished AKT phosphorylation but also reduced YAP phosphorylation (***Figure 4E***). As expected, upon AKTi administration concomitantly with APAP, the phosphorylation of both AKT and YAP decreased (***Figure 4F***).

Since our alternative PDE model predicted that the nuclear YAP phosphorylation is crucial for protein shuttling and its activity (***Figure 2C–E***), we hypothesized that AKT and YAP not only physically interact in the cytoplasm but also in cell nuclei. Indeed, PLA illustrated that AKT and YAP, as well as the respective phosphorylated isoforms, interacted in the nucleus (***Figure 4G,H***). Although, interaction between AKT/YAP and pAKT/pYAP was also detectable in the cytoplasm, a prominent protein colocalization took place in the nuclear compartment (***Figure 4G***). Importantly, image-based quantification revealed that the interaction between nuclear YAP and AKT increased upon 6hr treatment with APAP (***Figure 4I***), indicating an APAP-dependent shift to nuclear phosphorylation and interaction, as predicted by the PDE model. Indeed, blocking AKT activity by AKTi reduced YAP phosphorylation, moderately elevated nuclear YAP positivity, and increased the physical interaction between AKT and YAP in cell nuclei (***Figure 4F*** and data not shown).

In summary, our data show that APAP affects YAP activity and shuttling behavior by enhancing cellular ROS followed by nuclear AKT/YAP binding and YAP phosphorylation.

## Sequential activation of ROS, AKT, and Hippo/YAP in mouse livers after APAP intoxication

The results of our mathematical model (***Figure 2***) and the in vitro experiments (***Figure 3*** and ***Figure 4***) demonstrated a distinct sequence of molecular events that control YAP activity after APAP exposure: ROS induction is followed by AKT activation and YAP phosphorylation/inactivation (early events – up to 6 hr). This is followed by phase, where YAP is dephosphorylated and transcriptionally active (late events – 24 to 48 hr).

To confirm this in vivo, mice were injected with a hepatotoxic dose of APAP (300 mg/kg) and liver tissues were collected up to 16 days (in total at nine time points). Subsequently, liver specimens

were subjected to expression profiling and results were investigated regarding the presence of gene signatures specific for ROS, AKT, and Hippo/YAP activity (*De Marco et al., 2017*; *Han et al., 2008*; *Wang et al., 2018*). The experimental results showed that gene expression of ROS, AKT, and Hippo/ YAP target genes in livers was altered and prominently activated between 6 hr and 2 days after APAP treatment (*Figure 5A–C*). To compare the temporal dynamic of the gene signatures, $z$-scores of genes in the signature per time point were summarized and normalized to the number of genes in the signature. As indicated by the results from the cell culture experiments, the expression index illustrated a specific order of signature activation starting with ROS (6–12 hr), followed by AKT (12 hr to 1 day) and Hippo/YAP signature genes (1–2 days) (*Figure 5D*).

To confirm the findings from the gene expression analysis, we performed immunohistochemical staining of liver tissues isolated at five time points after APAP treatment. Staining for total YAP illustrated not only a general increase of YAP positivity in the cytoplasm of surviving hepatocytes, but also its prominent nuclear accumulation after 1–2 days (*Figure 5E*, arrows). Moreover, nuclear AKT enrichment was already detectable 6 hr after APAP treatment. Interestingly, nuclear AKT accumulation in hepatocytes remained high for the rest of the experiment compared to untreated mice (*Figure 5E*, arrows).

Because of a high mouse-to-mouse variability, we decided to objectify these results. For this, the immunohistochemical staining of YAP and AKT was quantitatively analyzed using machine learning (random forest) and image analysis methods (*Figure 5F*). For this, 733 (for YAP) and 516 (for AKT) images were unbiasedly selected from all investigated animals and analyzed with an algorithm, which was trained to detect positively stained nuclei outside of the necrotic areas and tissue borders (*Figure 5—figure supplement 1*; for detailed information see Materials and methods). The quantification results confirmed the observed nuclear YAP positivity 1–2 days after APAP administration, while nuclear AKT was already detectable at 6 hr and maintained during the experiment (*Figure 5F*). The shift from exclusive nuclear AKT to nuclear YAP was illustrated by direct comparison of the time points 6 hr and 2 days after APAP injection (*Figure 5G*). The mechanistic connection between AKT activity and YAP induction in murine hepatocytes was confirmed in independent experiments. Here, hydrodynamic gene delivery of myristoylated AKT led to nuclear enrichment of YAP expression in hepatocytes and subsequent induction of typical YAP target genes (data not shown).

To sum up, we confirm the sequential order of APAP-induced events that cause YAP activation in mouse liver tissue.

## Discussion

In this study, we aimed to investigate the different properties of the Hippo pathway effectors YAP and TAZ under in vitro and in vivo conditions that resembled the situation in DILI. For this, we developed a cell-based model system that allowed the systematic and comparative tracing of spatial YAP/TAZ dynamics upon stimulation with APAP, which is one of the most widely used analgesic with high potential of liver toxicity. By integrating time-resolved experimental data, computational modeling, image analysis tools as well as confirmatory in vivo results, we could draw several important conclusions on the role of YAP and TAZ under physiological conditions and in liver cells upon DILI.

Several mathematical modeling approaches have been previously applied to explain the dynamic behavior of the Hippo pathway under physiological or pathological conditions in different organisms. For example, computational modeling was used to explain how Hippo signaling cross-talks with other signaling pathways via protein interactions (*Labibi et al., 2020*; *Romano et al., 2014*). Other studies exclusively focused on biophysical/biochemical regulation of the Hippo pathway to investigate how YAP/TAZ activity relates to mechanical input information (*Ege et al., 2018*; *Eroumé et al., 2021*; *Gou et al., 2018*; *Scott et al., 2021*; *Sun et al., 2016*).

The goal of our approach was to determine whether nuclear phosphorylation of YAP plays a role in Hippo signaling, specifically in governing YAP localization. A model not including nuclear phosphorylation of YAP could not sufficiently explain the experimentally observed localization pattern and was excluded from further analyses. However, the alternative model proposed in this work, can explain the experimental data and therefore establishes a possible mechanism for YAP localization. However, more complex mechanisms regulating subcellular localization and dynamics of YAP and TAZ cannot be excluded. Therefore, our model partly explains how the Hippo pathway is organized, however, it does not aim to comprehensively describe it.

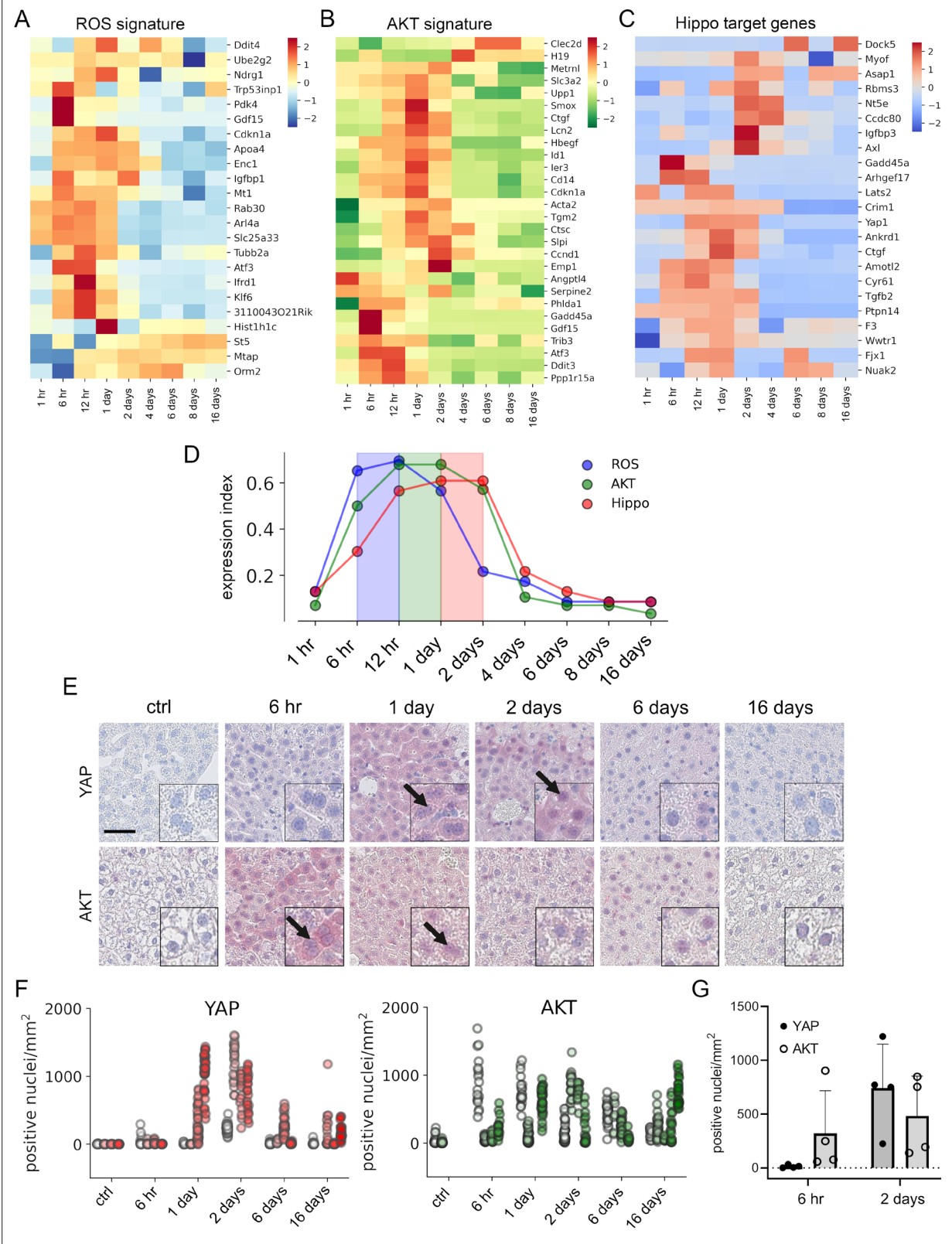

**Figure 5.** APAP stimulates the sequential activation of reactive oxygen species (ROS), AKT, and YAP in vivo. Gene expression profiling over time of mouse livers after APAP treatment (**A–C**) was performed for up to 16 days (3–5 animals/group, 300 mg/kg). Abundance of gene signatures characteristic for the activity of ROS consisting of 23 genes (*Han et al., 2008*) (A), AKT (**B**) with 28 genes (*De Marco et al., 2017*) and YAP (**C**) consisting of 23 genes (*Wang et al., 2018*) were analyzed. In A–C, gene expression values are z-score normalized. (**D**) Summarized gene expression scores (expression index)

*Figure 5 continued on next page*

*Figure 5 continued*

over time illustrate the timely order of ROS (max values between 6 and 12 hr), AKT (max values between 12 hr and 1 day) and YAP (max values between 1 and 2 days) target gene signatures after APAP treatment. (**E**) Mouse liver tissue sections stained for YAP (top tow) and total AKT (bottom row) under control condition and at 6 hr as well as 1, 2, 6, and 16 days after APAP treatment (300 mg/kg). Arrows indicate high YAP and AKT positivity in nuclei. Scale bar: 50 μm. (**F**) Automatic quantification of nuclei with YAP and AKT positivity in mouse liver tissues. Each dot represents the count of stained nuclei in one image (a tile, 1 mm$^2$) for YAP ($n$ = 733) and AKT ($n$ = 520). (**G**) Quantification of YAP and AKT nuclear positivity 6 hr and 2 days after APAP treatment. One dot represents the average count of positive nuclei in one animal ($n$ = 4).

The online version of this article includes the following figure supplement(s) for figure 5:

**Figure supplement 1.** Quantification of YAP- and AKT-positive nuclei using immunohistochemically stained tissue slides.

Here, we decided to use PDE modeling, which allowed us to discover spatial aspects of signaling dynamics and to use live cell image data for parametrizing the Hippo signaling steady-state models. By doing so, we compared PDE model simulations of different model topologies with the experimentally acquired data for YAP as well as TAZ and selected a model topology, which best corresponded to data generated in vitro. The selected alternative Hippo pathway model is based on experimental observations and parametrized within the considered boundaries of experimental evidence (see Appendix 1). The obtained model parameters are results of the best model fit obtained, and allow reproducing the described model behavior. However, since the model fitting is based on steady-state data only and concentration observations are in arbitrary units, the parameter values are subject to structural non-identifiability. Nevertheless, this does not affect the qualitative conclusions drawn from the models for the investigated biological process.

The results of our PDE modeling approach strongly suggested that the nuclear phosphorylation reaction of YAP/TAZ was necessary and sufficient for the reproduction of their subcellular distribution patterns in vitro (i.e., the halo-like nuclear distribution of YAP and TAZ). This observation is of importance since the conventional view on the Hippo pathway topology represents sequential phosphorylation steps in the cytoplasm. Our PDE modeling-based finding would therefore extend the current understanding of Hippo pathway regulation since it adds the cell nucleus as a pivotal spatial compartment in the modulation and adjustment of Hippo/YAP/TAZ signaling.

This conclusion is supported by our experimental findings illustrating that LATS1/2 are expressed in cell nuclei and that the interaction between phosphorylated LATS1/2 and YAP as well as between AKT and YAP is also detectable in this compartment. Importantly, previously published experimental and computational approaches support our findings. For example, nuclear LATS1/2 controls YAP phosphorylation in the context of NF2/Merlin deficiency (*Li et al., 2014*). In addition, the combination of quantitative photobleaching experiments and modeling illustrated that nuclear export is a crucial reaction for the subcellular YAP distribution (*Ege et al., 2018*). Combining these insights on nuclear export mechanisms with our findings on nuclear YAP phosphorylation strongly suggests that both processes closely cooperate in the efficient regulation of the Hippo pathway effectors. Lastly, our model does not explicitly exclude phosphorylation reaction in the cytoplasm; however, it suggests the necessity to reevaluate the canonical Hippo pathway scheme. The appeal for the reevaluation and extension of the canonical Hippo pathway, for example, by combining models on nuclear phosphorylation and nuclear transport, would further broaden our understanding on the dynamic spatial behavior of complex signaling pathways (*Ege et al., 2018*; *Shreberk-Shaked and Oren, 2019*).

The extension of the canonical Hippo pathway model with processes that control active nuclear YAP phosphorylation illustrates that the regulatory YAP/TAZ phosphorylation must be considered as continuum process (*Shreberk-Shaked and Oren, 2019*). For instance, the existence of nuclear phosphorylation possibilities by LATS-dependent and -independent (e.g., by AKT) mechanisms increases the complexity but also flexibility and redundancy of the signaling pathway (*Ege et al., 2018*; *Gao et al., 2017*; *Low et al., 2014*). These cellular aspects are part of a fine-tuned regulatory network that allows a rapid and adjustable proliferative response of YAP and TAZ under diverse physiological and pathological cellular conditions (e.g., upon induction of tissue regeneration).

To our knowledge, a direct mathematical comparison of YAP and TAZ regarding their spatial distribution has not been performed, yet. Using the PDE model, we comparatively investigated YAP and TAZ distribution and identified kinetic parameters, which discriminate between both species. Specifically, the observed localization differences (nuclear exclusion vs. nuclear localization) between YAP and TAZ might be caused by distinct phosphorylation rates of YAP and TAZ. Nuclear/cytoplasmic

shuttling dynamics for YAP and/or TAZ have been investigated previously (*Ege et al., 2018*; *Plouffe et al., 2018*). However, our model predicts that the phosphorylation efficiency for TAZ must be higher than for YAP to achieve the observed differential localization of YAP and TAZ.

By utilizing our image analysis approach for the detection of minor changes in the subcellular distribution of YAP and TAZ, we showed a dynamic shuttling response of YAP (and to a much lesser extent of TAZ) upon APAP treatment. Although, APAP-dependent activation of YAP was described in the literature (*Poudel et al., 2021*), the underlying molecular mechanisms were poorly understood. Our results demonstrated that APAP-induced YAP shuttling is not the result of an unspecific cellular response but is due to a sequential order of molecular events that include ROS induction (*Ghallab et al., 2016*; *Shuhendler et al., 2014*), followed by AKT (in)activation (*Koundouros NPoulogiannis, 2018*) and nuclear YAP (de)phosphorylation. Time-dependent APAP-induced ROS activation was recently supported by a study by Ghallab et al., which showed transient induction of oxidative stress in mice 8 hr after APAP overdose with a peak at 2 hr (*Ghallab et al., 2022*). Interestingly, we observed a biphasic YAP regulation: YAP phosphorylation/inactivation at early time points (up to 3 hr) and its dephosphorylated/activation at later time points (>24 hr). Especially, the functionally important activation of YAP, which is associated with the induction of cell proliferation, is well reflected by the temporal activation of YAP in murine tissues upon APAP administration after 24–48 hr.

Importantly, several cellular mechanisms might simultaneously contribute to the APAP-dependent activation of YAP. For example, the direct impact of APAP on MST1/2 kinases, which are central Hippo pathway constituents, represents one additional process that may contribute to the observed effects on YAP (*Fan et al., 2016*). Moreover, the serine-/threonine-kinase AKT could control the Hippo/YAP axis at different levels. For example, a physical interaction of AKT and YAP in the nucleus (shown here) or the cytoplasm might directly control YAP phosphorylation (*Basu et al., 2003*). Alternatively, AKT can phosphorylate MST2 and therefore indirectly contribute to YAP activity (*Romano et al., 2014*). Thus, APAP probably controls YAP in a multimodal manner to achieve a cellular response and it is therefore likely that this multimodal process is part of cell-protective mechanism that counteracts the massive loss of hepatocytes upon APAP-induced DILI (*Holland et al., 2022*; *Shuhendler et al., 2014*).

In our manuscript, we show that APAP administration induces a chain of molecular events through the APAP/ROS/AKT axis, which is leading to YAP inactivation (early) followed by YAP activation (late). The late induction of YAP activity might be considered as cell-protective cellular response upon long-term tissue damage; however, YAP also acts as a potent oncogene in different cell types, including hepatocytes, and its overexpression is associated with tumor initiation (*Dong et al., 2007*; *Weiler et al., 2017*). Thus, it is tempting to speculate that APAP overdose may not only cause DILI but also increases the risk for liver tumor development. Although controversially discussed in the literature, a recent evaluation of 139 published epidemiologic studies strongly argues against a relationship between APAP exposure and cancer (*Weinstein et al., 2021*). Actually, APAP has been discussed as therapeutic option for patients with YAP-induced cancer (*Poudel et al., 2021*). Although, our and other studies clearly demonstrate a mechanistic link between APAP uptake and Hippo/YAP pathway activity, these partly controversial findings illustrate the necessity to decipher this connection under distinct disease conditions in the future.

## Materials and methods

### Key resources table

| Reagent type (species) or resource | Designation | Source or reference | Identifiers | Additional information |
|---|---|---|---|---|
| Gene (*Homo sapiens*) | *YAP* | Prof. Loewer (TU Darmstadt) | ENSG00000137693 | – |
| Gene (*Homo sapiens*) | *TAZ* | Cloned | ENSG00000018408 | – |
| Cell line (*Homo sapiens*) | Hep3B | DSMZ | #ACC93 | – |
| Cell line (*Homo sapiens*) | HepG2 | LGC | ATCC-HB-8065 | – |

*Continued on next page*

*Continued*

| Reagent type (species) or resource | Designation | Source or reference | Identifiers | Additional information |
|---|---|---|---|---|
| Cell line (*Homo sapiens*) | HLF | JCRB | JCRB0405 | – |
| Cell line (*Homo sapiens*) | SNU182 | ATCC | CRL-2235 | – |
| Transfected construct | pRRLN-EF1α-mVenus-YAP | Prof. Loewer (TU Darmstadt) | End-to-end sequencing | – |
| Transfected construct | pRRLN-EF1α-d2mCherry-TAZ | This paper | End-to-end sequencing | – |
| Transfected construct | pLentiPGK-mCerulean-H2B | Addgene | 90234 | – |
| Chemical compound, drug | AKT inhibitor VIII | Merck Millipore | 124018 | – |
| Sequence-based reagent | siRNA YAP#1 | This paper | | CCACCAAGCUAG AUAAAGA-dT-dT |
| Sequence-based reagent | siRNA YAP#2 | This paper | | GGUCAGAGAUAC UUCUUAA-dT-dT |
| Sequence-based reagent | control siRNA | This paper | | UGGUUUACAUG UCGACUAA |
| Antibody | YAP (rabbit monoclonal) | Cell Signaling Technology | RRID:AB_2650491 | WB (1:1000) |
| Antibody | pYAP (rabbit polyclonal) | Cell Signaling Technology | RRID:AB_2218913 | WB (1:400) PLA (1:200) |
| Antibody | pAKT (rabbit monoclonal) | Cell Signaling Technology | RRID:AB_2315049 | WB (1:1000) |
| Antibody | AKT (rabbit polyclonal) | Cell Signaling Technology | RRID:AB_329827 | WB (1:1000) PLA (1:200) |
| Antibody | β-Actin (mouse monoclonal) | MP Biomedicals, Solon | RRID:AB_2335127 | WB (1:10,000) |
| Antibody | LATS1 (mouse monoclonal) | Santa Cruz Biotechnology | sc-398560 | WB (1:200) |
| Antibody | LATS2 (mouse monoclonal) | Santa Cruz Biotechnology | sc-515579 | WB (1:200) |
| Antibody | pLATS1/2 (rabbit monoclonal) | Cell Signaling Technology | RRID:AB_10971635 | WB (1:500) PLA (1:200) |
| Antibody | CYP2E1 (rabbit unknown) | Novus Biologicals | RRID:AB_11021447 | WB (1:500) |
| Antibody | PARP (rabbit polyclonal) | Cell Signaling Technology | RRID:AB_2160739 | WB (1:10,000) |
| Antibody | β-Tubulin (mouse monoclonal) | Santa Cruz Biotechnology | RRID:AB_2288090 | WB (1:200) |
| Antibody | GAPDH (chicken polyclonal) | Merck Millipore | RRID:AB_10615768 | WB (1:10,000) |
| Antibody | YAP (mouse monoclonal) | Santa Cruz Biotechnology | RRID:AB_10612397 | PLA (1:25) |
| Antibody | pAKT (mouse monoclonal) | Cell Signaling Technology | RRID:AB_331158 | PLA (1:200) |
| Antibody | YAP (rabbit monoclonal) | Cell Signaling Technology | RRID:AB_2650491 | IHC (1:50) |
| Antibody | AKT (rabbit polyclonal) | Cell Signaling Technology | RRID:AB_329827 | IHC (1:50) |
| Commercial assay or kit | ROS assay kit | Abcam | ab287839 | – |
| Sequence-based reagent | YAP (human) – forward | This paper | NM_006106 | CCTGCGTAGCCAGTTACCAA |
| Sequence-based reagent | YAP (human) – reverse | This paper | NM_006106 | CCATCTCATCCACACTGTTC |
| Sequence-based reagent | ANKRD1 (human) – for | This paper | NM_014391.3 | AGTAGAGGAACTGGTCACTGG |
| Sequence-based reagent | ANKTD1 (human) – rev | This paper | NM_014391.3 | TGGGCTAGAAGT GTCTTCAGA T |

*Continued on next page*

*Continued*

| Reagent type (species) or resource | Designation | Source or reference | Identifiers | Additional information |
|---|---|---|---|---|
| Sequence-based reagent | CYR61 (human) – for | This paper | NM_001554.5 | AGCCTCGCATCCTATACAACC |
| Sequence-based reagent | CYR61 (human) – rev | This paper | NM_001554.5 | TTCTTTCACAAGGCGGCACTC |
| Sequence-based reagent | GAPDH (human) – for | This paper | NM_002046.7 | CTGGTAAAGTGGATATTGTTGCCAT |
| Sequence-based reagent | GAPDH (human) – rev | This paper | NM_002046.7 | TGGAATCATATTGGAACATGTAAACC |
| Sequence-based reagent | RPL41 (human) – for | This paper | NM_001035267 | AAACCTCTGCGCCATGAGAG |
| Sequence-based reagent | RPL41 (human) – rev | This paper | NM_001035267 | AGCGTCTGGCATTCCATGTT |
| Sequence-based reagent | SRDF4 (human) – for | This paper | NM_005626 | TGCAGCTGGCAAGACCTAAA |
| Sequence-based reagent | SRSF4 (human) – rev | This paper | NM_005626 | TTTTTGCGTCCCTTGTGAGC |
| Sequence-based reagent | B2M (human) – for | This paper | NM_004048 | CACGTCATCCAGCAGAGAAT |
| Sequence-based reagent | B2M (human) – rev | This paper | NM_004048 | TGCTGCTTACATGTCTCGAT |
| Software, algorithm | ASAP software | https://github.com/computationalpathologygroup/ASAP | v1.6 | – |
| Software, algorithm | ImageJ | *Rueden et al., 2017* | v1.53f51 | – |
| Software, algorithm | Weka segmentation | *Arganda-Carreras et al., 2017* | v3.3.1 | – |
| Software, algorithm | Ilastik | *Berg et al., 2019* | v1.3.3 | – |
| Software, algorithm | Spatial Model Editor | https://spatial-model-editor.github.io | v1.2.1 | – |
| Software, algorithm | sme-contrib | https://spatial-model-editor.github.io | v0.014 | – |

## In vitro experiments

### Cell culture and the establishment of genetically modified cells

The hepatocyte-derived cell lines Hep3B (#ACC93, DSMZ, Braunschweig, Germany), HLF (JCRB, Japan), HepG2, and SNU182 (ATCC/LGC Standards, Wesel, Germany) were cultured in Minimum Essential Medium (MEM), Dulbeccos Modified Eagle Medium (DMEM), and Roswell Park Memorial Institute (RPMI) medium, respectively. The media were supplemented with 10% FCS and 1% peni-cillin–streptomycin (Sigma-Aldrich, Taufkirchen, Germany). Cells were maintained at 37°C in a 5% $CO_2$ atmosphere. Hep3B cells were transduced with vectors carrying cDNAs for human histone *H2B*, *YAP*, and *TAZ/WWTR1* genes, fused with mCerulean, mVenus, and mCherry genes, respectively. The plasmid pRRLN-EF1α-mVenus-YAP was kindly provided by Prof. Dr. Alexander Loewer (TU Darmstadt), pLentiPGK-mCerulean-H2B was purchased (Addgene, Watertown, USA; No. 90234). The pRRLN-EF1α-d2mCherry-TAZ was cloned. Correctness of vectors was verified by end-to-end sequencing.

Vectors were stably integrated into the Hep3B cells using lentiviral particles. For this, HEK293 cells were transfected with lentiviral vectors, packaging (psPAX2, Addgene, Watertown, USA; No. 12260), and envelope plasmids (pMD2.G, Addgene, Watertown, USA; No. 12259) using polyethylenimine and incubated for 40 hr. Subsequently, virus particles in the cell culture medium were collected and used for infections. The transduced cells were selected for stable vector integration using geneticin (0.6 mg/ml, pRRLN-EF1α-mVenus-YAP), hygromycin (0.6 mg/ml, pLentiPGK-mCerulean-H2B), and

blasticidin (2 µg/ml, pRRLN-EF1α-d2mCherry-TAZ). Cells were frequently checked for mycoplasma contamination and authenticated by short tandem repeat analysis (DSMZ, Braunschweig, Germany).

The APAP stock solution (100 mM) was freshly prepared by dissolving 0.3 g APAP crystalline powder in 20 ml phosphate-buffered saline (PBS, GE Healthcare, Solingen, Germany) under heating (42°C) and stirring (Sigma-Aldrich, St. Louis, USA). The APAP solution was filtered (Millex-GS filters, 0.22 µm; Merck Millipore Ltd, Carrigtwohill, Ireland) and immediately administered to cells to a final concentration of 10 mM. TBHP (Luperax, Sigma-Aldrich) with 70 wt% in $H_2O$ was applied to cells to a final concentration of 300 µM for 2 hr prior protein extraction. Hydrogen peroxide ($H_2O_2$) 30% (Roti-puran, Carl Roth GmbH, Karlsruhe, Germany) was applied to a final concentration of 2 mM for 1 hr.

AKT1/2 phosphorylation was inhibited with InSolution Akt Inhibitor VIII (1–10 µM, Merck KGaA, Darmstadt, Germany) in FCS-free medium for 3 hr prior to treatment with HGF (10 ng/µl).

## Gene silencing by RNAi

RNAi was performed to inhibit *YAP* gene expression. For this, $2 \times 10^5$ cells were seeded per well in a 6-cm-well plate and incubated overnight. Oligofectamine (Invitrogen, Carlsbad, CA, USA) and Opti-MEM (Gibco Life Technologies Ltd, Paisley, UK) were used for transfection of the siRNA (final concentration: 5 nM) according to the manufacturer's instructions. A random sequence oligonucleotide was used as a control treatment (control siRNA: ctrl siRNA). The following siRNA sequences were used: YAP siRNA #1 (5′–3′): CCA CCA AGC UAG AUA AAG A-dT-dT and YAP siRNA #2 (5′–3′): GG UCA GAG AUA CUU CUU AA-dT-dT, and ctrl siRNA (5′–3′): UGG UUU ACA UGU CGA CUA A. YAP siRNA #1 and #2 were pooled to achieve an optimal gene knockdown. Twenty-four hours after siRNA transfection, cells were treated with APAP (10 mM) for 24 hr.

## Live cell imaging

Stably transduced Hep3B cells were seeded on 24-well glass-bottom black wall plates (MoBiTec GmbH, Goettingen, Germany) in a phenol red-free RPMI 1640 medium (Gibco/Life Technologies Corporation, Paisley, UK). Dynamic protein localization in living cells was measured using confocal laser scanning microscope (Nikon A1R on an inverted Nikon Ti2 microscope), which was supplemented with an on-stage incubator (TokaiHit) to maintain 37°C and 5% $CO_2$ at all stages of the experiments. Selected images were acquired with additional Nikon AxR microscope system.

Three lasers were employed: 445 nm (mCerulean, PMT detector), 514 nm (mVenus, GaAsP detector), and 561 nm (mCherry, GaAsP detector) to obtain highly resolved live cell images. Two channels – 445 and 514 nm – were acquired simultaneously. The employed pinhole size was 17.9 µm (445 nm: 1.5 a.u., 514 nm: 1.3 a.u., 561 nm: 1.2 a.u.). Images were acquired using a galvano-scanner of size 512 × 512 px (FOV 0.64 × 0.64 mm, pixel resolution of 1.24 µm). The objectives Plan Apo $\lambda$ 20× (NA 0.75, working distance 1 mm, FOV 0.64 × 0.64 mm) or Plan Apo $\lambda$ 40× (NA 0.95, working distance 0.21 mm, FOV 0.435 × 0.435 mm) were used.

## Western immunoblotting

Western immunoblotting was performed as previously described (*Weiler et al., 2017*). In brief, for total protein fraction preparation, cultured cells were harvested using 1× Cell Lysis Buffer (Cell Signaling Technology, Danvers, USA) supplemented with 1× Protease Inhibitor Mix G (Serva, Heidelberg, Germany) and 1× PhosSTOP (Roche Diagnostics Deutschland GmbH, Mannheim, Germany). For protein fractionation, the NE-PER Nuclear and Cytoplasmic Extraction kit was used (Thermo Scientific, Rockford, USA). The following antibodies were used for western immunoblotting experiments: YAP XP (1:1000, Cell Signaling Technology, Danvers, USA, #14074, RRID:AB_2650491), pYAP (1:400, Cell Signaling Technology, #4911, RRID:AB_2218913), pAKT (1:1'000, Cell Signaling Technology, #4060, RRID:AB_2315049), AKT (1:1'000, Cell Signaling Technology, #9272, RRID:AB_329827), β-actin (1:10,000, MP Biomedicals, Solon, USA, #08691001, RRID:AB_2335127), LATS1 (1:200, Santa Cruz Biotechnology, Dallas, USA, sc-398560), LATS2 (1:200, Santa Cruz Biotechnology, sc-515579), and pLATS1/2 (1:500, Cell Signaling Technology, #8654, RRID:AB_10971635) and Cytochrome P450 2E1 (1:500, Novus Biologicals, Centennial, USA, NVP1-85367, RRID:AB_11021447). PARP (1:10,000, Cell Signaling Technology, #9542, RRID:AB_2160739), β-tubulin (1:200, Santa Cruz Biotechnology, sc-5274, RRID:AB_2288090), and GAPDH (1:10,000, Merck Millipore, Darmstadt, Germany, ab2302, RRID:AB_10615768) were used as loading controls.

Western blot detection and quantification were performed using the Odyssey-CLx Infrared Imaging system with the ImageStudio software (LI-COR Biosciences, Bad Homburg, Germany). Phosphorylated protein band intensity was measured and normalized to total protein concentrations. Equal amount of protein was loaded for each line, as measured by the Bradford reagent (Millipore Sigma, Saint Louis, USA). Raw unedited image files and uncropped blots are available as source data files of this manuscript.

## RNA isolation, reverse transcription, and quantitative real-time PCR (qPCR)

RNA was isolated using Extractme kit (Blirt, Gdańsk, Poland) according to the manufacturer's protocol. Reverse transcription was performed using the RevertAid kit (Thermo Scientific). Semiquantitative real-time PCR reactions were set up using the primaQuant 2× qPCR-SYBR-Green-Mastermix (Steinbrenner Laborsysteme, Wiesenbach, Germany) and analyzed with the QuantStudio 3 real-time PCR system (Applied Biosystems, Thermo Fisher Scientific, Singapore). The following cycling conditions were applied: 95°C for 15 min, followed by 40 cycles of 95°C for 15 s, and 60°C for 60 s. Product specificity was confirmed by melting curve analysis (95°C for 15 s, 60°C for 30 s, 60–95°C with 0.5°C/s).

The mRNA levels were normalized to glyceraldehyde-3-phosphate dehydrogenase (*GAPDH*), 60S ribosomal protein L41 (*RPL41*), serine/arginine-rich splicing factor (*SRSF4*), and β2-microglobulin (*B2M*). The following primers for human cDNAs were used: YAP-for: 5′-CCT GCG TAG CCA GTT ACC AA-3′; YAP-rev: 5′-CCA TCT CAT CCA CAC TGT TC-3′; ANKRD1-for: 5′-AGT AGA GGA ACT GGT C AC TGG-3′; ANKRD1-rev: 5′-TGG GCT AGA AGT GTC TTC AGA T-3′; CYR61-for: 5′-AGC CTC GCA TCC TAT ACA ACC-3′; CYR61-rev: 5′-TTC TTT CAC AAG GCG GCA CTC-3′; GADPH-for: 5′-CTG G TA AAG TGG ATA TTG TTG CCA T-3′; GAPDH-rev: 5′-TGG AAT CAT ATT GGA ACA TGT AAA CC-3′; RPL41-for: 5′-AAA CCT CTG CGC CAT GAG AG-3′; RPL41-rev: 5′-AGC GTC TGG CAT CCA TGT TT-3′; SRSF4-for: 5′-TGC AGC TGG CAA GAC CTA AA-3′; SRSF4-rev: 5′-TTT TTG CGT CCC TTG TGA GC-3′; B2M-for: 5′-CAC GTC ATC CAG CAG AGA AT-3′; B2M-rev: 5′-TGC TGC TTA CAT GTC TCG AT-3′. For the analysis of gene expression in tissue samples, a panel of housekeeping genes was analyzed using the geNorm algorithm to find the most stable reference gene (*Vandesompele et al., 2002*).

## In situ PLA

The DuoLink in situ PLA was performed according to the manufacturer's instructions (Sigma-Aldrich). Briefly, cells were seeded on glass coverslips and prior to APAP administration cells were grown under FCS-free conditions for 1 day, then incubated with APAP (10 mM) or PBS. After treatment, cells were washed three times with 2 mM MgCl$_2$ in PBS and fixed with 4% paraformaldehyde for 10 min at room temperature. Fixed cells were washed four times with PBS for 5 min, permeabilized with 0.2% Triton X-100 in PBS for 5 min at room temperature, and washed again twice with PBS for 5 min. Subsequently, cells were blocked with Blocking solution (Sigma-Aldrich) for 30 min at room temperature and incubated with the following primary antibodies diluted in Antibody Diluent (Sigma-Aldrich) overnight at 4°C: anti-YAP (1:25, Santa Cruz Biotechnology, sc-271134, RRID:AB_10612397), anti-AKT (1:200, Cell Signaling Technology, #9272, RRID:AB_329827), anti-pYAP (1:200, Cell Signaling Technology, #4911, RRID:AB_2218913), anti-pAKT (1:200, Cell Signaling Technology, #4051, AB_331158), and anti-pLATS1/2 (1:200, Cell Signaling Technology, #8654, RRID:AB_10971635). Subsequently, cells were washed twice with Wash Buffer A (Sigma-Aldrich) and incubated with prediluted rabbit PLUS and mouse MINUS probes (Sigma-Aldrich) in Antibody Diluent for 1 hr at 37°C. After incubation, cells were washed twice with Wash Buffer A and then incubated with ligation solution (Sigma-Aldrich) for 30 min at 37°C. After ligation, samples were again washed twice with Wash Buffer A for 2 min at room temperature and incubated with amplification solution, and detection reagent Orange (Sigma-Aldrich) for 100 min at 37 °C. Finally, cells were washed twice with Wash Buffer B (Sigma-Aldrich) for 10 min at room temperature, once with 0.01× Wash Buffer B for 1 min at room temperature and coverslips were mounted on the slide with DAPI Fluoromount-G mounting medium (SouthernBiotech, Birmingham, USA).

Fluorescence images were captured using an inverted Nikon Ti2 microscope with Nikon S Plan Fluor ELWD ×40 NA 0.60 objective in a widefield fluorescence mode using Lumencor Sola SE II lamp. Images were captured in DAPI and TRITC channels (460 and 580 nm) with Nikon DS-Qi2 monochrome camera (image size 2404 × 2404 px, pixel resolution of 0.18 µm/px).

## ROS activity measurement

Measurements of the intracellular ROS levels were performed using the DCFDA/H2DCFDA cellular ROS assay kit (Abcam, Amsterdam, Netherlands) according to the manufacturer's protocol. In brief, cells were seeded on white clear-bottom 96-well plate (Corning, Corning, USA) and incubated overnight. Cells were treated with 10 mM APAP, 2 mM $H_2O_2$, or 300 µM TBHP ($H_2O_2$ and TBHP served as positive controls for ROS induction) for 6 hr in FCS-free cell culture medium. Fluorescence was measured using a microplate reader (FluoStar Omega, BMG Labtech GmbH, Ortenberg, Germany). Buffer solution without cells served as a background control.

## In vivo experiments and sample analyses

### Housing and treatment of mice and induction of acute liver injury by acetaminophen

Male C57BL/6N mice (8- to 10-week-old) were bought from Janvier Labs (Janvier Labs, Le Genest-Saint-Isle, France). Animals were housed under 12 hr light/dark cycles at controlled ambient temperature of 25°C with free access to water and were fed ad libitum with a standard diet (Ssniff, Soest, Germany) before starting the experiments. Induction of acute liver injury with APAP was done as previously described (*Schneider et al., 2021*). Briefly, the mice were fasted overnight, then challenged with a dose of 300 mg/kg APAP intraperitoneally. APAP was dissolved in warm PBS with an application volume of 30 ml/kg. Control group was treated with PBS only. The mice were fed ad libitum after APAP administration. All animals were included for further analyses. All experiments were approved by the local animal welfare committee (LANUV, North Rhine-Westphalia, Germany, application number: 84-02.04.2016.A279). Hydrodynamic gene delivery experiments in mice were performed as recently described (*Luiken et al., 2020*).

### Liver tissue sample collection, processing, and staining

Tissues were collected time dependently after APAP injection from the left liver lobe. The tissues were fixed and embedded in paraffin as previously described (*Ghallab et al., 2016*). YAP and AKT immunostaining were performed using 4-µm-thick paraffin-embedded tissue sections. For immuno-histochemistry, an anti-YAP antibody (1:50, Cell Signaling Technology, #14074, RRID:AB_2650491) and anti-AKT (1:50, Cell Signaling Technology, #9272, RRID:AB_329827) were used. Embedded tissue sections were pretreated with a heat-induced epitope retrieval method (pH 6, DAKO, Hamburg, Germany). As secondary antibody anti-rabbit Polymer-AP (Enzo Life Sciences, Farmingdale, USA, ENZ-ACC110-0150) was used. Detection was performed with Permanent AP (Zytomed Systems GmbH, Berlin, Germany). Following staining, the whole slides were digitally documented using a slide scanner (Aperio AT2, Leica Mikrosysteme Vertrieb GmbH, Wetzlar, Germany).

### Expression profiling and bioinformatics

RNA isolation and gene array analysis were performed as published before (*Campos et al., 2020*) and bioinformatic analysis was done as described (*Holland et al., 2022*). The gene expression data are available under ArrayExpress accession number GSE167032. The expression data were applied to three known signatures that are informative for ROS activity (*Han et al., 2008*), AKT activity (*De Marco et al., 2017*), and YAP/TAZ activity (*Wang et al., 2018*). Due to the high number of AKT signature genes, only genes whose response to APAP treatment was larger than fold change of 2 (compared to control animals) were considered. The expression data were z-score normalized and clustered with seborn clustermap python module (v0.11.0). The signature score was obtained by summarizing z-scored expression values at the given time point if the z-scored value was greater than 0.5. The summarized expression values were normalized to the number of genes in the signature.

## Computational methods

### Analysis of the live cell images

The confocal images of the living cells were analyzed in a high-throughput manner using ImageJ (v1.53f51) platform (*Rueden et al., 2017*). Images were first manually selected with respect to the quality criteria: images with insufficient sharpness or with artifacts were discarded from the analysis. The image processing pipeline was based on Weka segmentation (v3.3.1) (*Arganda-Carreras*

*et al., 2017*) of foreground and background areas, and subsequent thresholding, object detection, and counting. After object detection and counting, the respective masks were overlaid on the initial images to acquire YAP and TAZ intensity values for nuclei and cytoplasm. Mean pixel intensity from nuclear areas of the cells was divided by the mean pixel intensity of cytoplasmic regions, thus obtaining an NCR.

## Analysis of PLA images

PLA slides were analyzed using ImageJ (v1.53f51). First, nuclei and dots were classified using the Weka segmentation algorithm (v3.3.1) (*Arganda-Carreras et al., 2017*), thresholded, and counted. The pseudo-cytoplasmic (ring-shaped) area was created using the ImageJ's binary mask option dilate for 30 iterations on nuclear masks to obtain nuclear and cytoplasmic area for a comparative analysis. The thresholded nuclei or cytoplasmic masks were overlaid with detected dots to obtain the information on the subcellular localization of the protein interaction.

## Analysis of IHC images

Stained tissue samples were digitalized using Aperio slide scanner with ×40 magnification and pixel resolution of 0.253 μm/px (Aperio AT2, Leica Mikrosysteme Vertrieb GmbH, Wetzlar, Germany). The selected time points (6 hr and 1, 2, 6, 16 days) and control treatment were quantified using a pipeline, which consisted of python scripts (modules PIL v5.3.0, matplotlib v2.2.3), ASAP software (v1.6, https://github.com/computationalpathologygroup/ASAP) and Ilastik software (v1.3.3) (*Berg et al., 2019*).

First, digital images of the tissue sections were divided in tiles (1 mm$^2$, 400 × 400 pixels), which were binned (to reduce processing time and storage load) using ASAP and python modules. Some images were excluded due to staining and/or scanning artifacts. Tiles, which displayed tissue for at least 50% of their area, were kept for further processing. Machine-learning model, which was based on a random forest algorithm, was trained on a selected set of training tiles for YAP or AKT using Ilastik software. The algorithm was trained to detect positively stained nuclei, excluding necrotic areas and staining artifacts. Ilastik software was further used to export probability maps, which were possessed with ImageJ. In ImageJ, probability maps were thresholded and positive nuclei were counted and normalized to the area of the tile, which was occupied by the tissue.

## PDE modeling

Spatial modeling aimed at describing the variations in space of fluorescently labeled YAP and TAZ distribution within living Hep3B cells. The mathematical model is based on a system of PDEs. PDE modeling and parameter estimation were performed with the Spatial Model Editor (SME) software (v1.2.1) and sme-contrib (v0.0.14) python module (https://spatial-model-editor.github.io/). SME is graphical user interface-based model editing and simulation software compatible with systems biology markup language (SBML) standards. Models were simulated using simple Forward Time Centered Space (FTCS) solver.

In the demonstrated models, all reactions were defined by first-order kinetics (for PDEs see Appendix 1). For the parameter estimation, the model behavior was evaluated at steady state, that is the model was simulated until the concentration distribution did not change over time. Parameter estimation was performed using the particle swarm algorithm (20 particles, 200 iterations) to minimize a cost function consisting of the weighted sum of two terms: the squared per pixel differences between model and the target image, and the sum of squares of species concentration rates of change. In total 200 fitted parametrizations for each tested model were generated. The parameter space for the optimization algorithm was defined based on published data (Appendix 1). The parameter ranges of our canonical and alternative models were selected in a way to describe biologically meaningful value ranges and to avoid numeric instability during PDE simulations of the canonical model. The mathematical models were uploaded to the BioModels repository under the model identifier number MODEL2202080001. PDE model equations and parameters can be found in Supplementary Information (*Appendix 1—Tables 1–6*; *Ege et al., 2018*; *Jack et al., 1990*). Tables showing all fitted parameters for YAP and TAZ are provided (*Supplementary file 1*, *Supplementary file 2*).

## Statistics

Statistical analysis was performed using GraphPad Prism 9.2.0. Statistical tests are indicated in figure legends. Error bars depict standard deviation. Significance levels are as follows: $*p \leq 0.05$, $**p \leq 0.01$, $***p \leq 0.001$.

# Acknowledgements

We want to thank Patrizia Birner and Michaela Bissinger for their excellent technical support. In addition, we thank the Nikon Imaging Center at Heidelberg University (BioQuant), especially Dr. Ulrike Engel and Dr. Christian Ackermann, for supporting confocal imaging experiments. We also acknowledge the LSDF2 (URZ) for data storage. We thank Prof. Dr. Alexander Loewer (TU Darmstadt) for providing the YAP-mVenus vector and the Center for Model System and Comparative Pathology (CMCP, Heidelberg) for staining mouse tissue samples, especially Heike Conrad, Christine Schmitt, and Sarah Lammer. Tissue scanning was supported by the Tissue Bank of the National Center of Tumor Diseases (NCT, Heidelberg).

# Additional information

### Funding

| Funder | Grant reference number | Author |
|---|---|---|
| German Federal Ministry of Education and Research | 031L0074H | Kai Breuhahn |
| Deutsche Forschungsgemeinschaft | 505755359 | Kai Breuhahn |
| Deutsche Forschungsgemeinschaft | 314905040 | Kai Breuhahn |
| German Federal Ministry of Education and Research | 031L0158 | Liam Keegan Ursula Kummer Sven Sahle |
| Heidelberg University | Research training group "Mathematical Modeling for the Quantitative Biosciences (MMQB)" | Lilija Wehling |
| Deutsche Forschungsgemeinschaft | GH 276 | Ahmed Ghallab |
| Ministerio de Ciencia e Innovación | Programa Estatal de Promoción del Talento y su Empleabilidad (FPU17/01995) | Paula Fernández-Palanca |

The funders had no role in study design, data collection, and interpretation, or the decision to submit the work for publication.

### Author contributions

Lilija Wehling, Conceptualization, Data curation, Validation, Investigation, Visualization, Methodology, Writing – original draft, Writing – review and editing; Liam Keegan, Software, Writing – review and editing; Paula Fernández-Palanca, Validation, Investigation, Methodology, Writing – review and editing; Reham Hassan, Resources, Investigation, Writing – review and editing; Ahmed Ghallab, Resources, Investigation, Project administration, Writing – review and editing; Jennifer Schmitt, Yingyue Tang, Maxime Le Marois, Investigation, Writing – review and editing; Stephanie Roessler, Formal analysis, Investigation, Writing – review and editing; Peter Schirmacher, Conceptualization, Resources, Funding acquisition, Writing – review and editing; Ursula Kummer, Conceptualization, Resources, Project administration, Writing – review and editing; Jan G Hengstler, Resources, Project administration, Writing – review and editing; Sven Sahle, Resources, Software, Supervision, Funding acquisition, Writing – original draft, Project administration, Writing – review and editing; Kai Breuhahn,

Conceptualization, Resources, Supervision, Funding acquisition, Methodology, Writing – original draft, Project administration, Writing – review and editing

**Author ORCIDs**
Lilija Wehling ⓘ http://orcid.org/0000-0002-8697-5348
Reham Hassan ⓘ http://orcid.org/0000-0002-6569-7676
Ahmed Ghallab ⓘ http://orcid.org/0000-0003-0695-3403
Sven Sahle ⓘ http://orcid.org/0000-0002-5458-7404
Kai Breuhahn ⓘ http://orcid.org/0000-0002-2462-1229

**Ethics**
This study was performed according to animal welfare committee of The Ministry for Environment, Agriculture, Conservation and Consumer Protection of the State of North Rhine-Westphalia (LANUV, North Rhine-Westphalia, Germany, application number: 84-02.04.2016.A279).

**Decision letter and Author response**
Decision letter https://doi.org/10.7554/eLife.78540.sa1
Author response https://doi.org/10.7554/eLife.78540.sa2

## Additional files

**Supplementary files**
• Supplementary file 1. Fitted parameters for YAP.
• Supplementary file 2. Fitted parameters for TAZ.
• MDAR checklist

### Data availability

The mathematical models generated and analyzed during the current study are available in the BioModels repository (https://www.ebi.ac.uk/biomodels/) under accession number MODEL2202080001. The PDE simulation software is available over Github (https://spatial-model-editor.github.io/). The gene expression data analyzed during the current study are available in the Gene Expression Omnibus (GEO) repository under accession number GSE167032.

The following dataset was generated:

| Author(s) | Year | Dataset title | Dataset URL | Database and Identifier |
|---|---|---|---|---|
| Wehling L, Sahle S | 2022 | MODEL2202080001 | https://www.ebi.ac.uk/biomodels/MODEL2202080001 | EMBL-EBI BioModels, MODEL2202080001 |

The following previously published dataset was used:

| Author(s) | Year | Dataset title | Dataset URL | Database and Identifier |
|---|---|---|---|---|
| Ghallab A, Hengstler JG, Holland CH | 2021 | Expression data of the livers of male C57Bl6/N mice after i.p. injection of paracetamol (APAP) | https://www.ncbi.nlm.nih.gov/geo/query/acc.cgi?acc=GSE167032 | NCBI Gene Expression Omnibus, GSE167032 |

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

# Appendix 1

## PDE model equations

In the computational model, the equations which are simulated in cytoplasm or nucleus follow two-dimensional reaction diffusion equation. Initial conditions for the PDE are arbitrary since only steady-state solutions are considered. The equations in TAZ model are equivalent to YAP model equations.

Here, described in general terms as follows:

$$\frac{\partial C_s}{\partial t} = R_s + D_s \nabla^2 C_s$$

where
- $C_s$ is the concentration of species $S$ at position $(x, y)$ and time $t$
- $R_s$ is the reaction term for species $S$
- $D_s$ is the diffusion constant for species $S$
- $\nabla^2 C_s$ is the Laplacian of species $S$, which can be rewritten as

$$\nabla^2 C_s = \nabla \cdot \nabla C_s = \frac{\partial^2 C_s}{\partial x^2} + \frac{\partial^2 C_s}{\partial y^2}$$

The PDEs for the dynamics of YAPn/pYAPn (nuclear fraction) and YAP/pYAP (cytosolic fraction) proteins in the alternative model in the nucleus or cytoplasm are presented as follows.

$$\frac{\partial [YAP]}{\partial t} = R_1 + D_{YAP} \nabla^2 [YAP]$$

$$\frac{\partial [pYAP]}{\partial t} = -R_2 + D_{pYAP} \nabla^2 [pYAP]$$

$$\frac{\partial [YAPn]}{\partial t} = -R_5 + R_6 + D_{YAPn} \nabla^2 [YAPn]$$

$$\frac{\partial [pYAPn]}{\partial t} = R_3 - R_4 + D_{pYAPn} \nabla^2 [pYAPn]$$

where reaction rates $R$ are defined as follows:
$R_1$ as translation:

$$R_1 = K_{translation}$$

$R_2$ as degradation:

$$R_2 = K_{degradation} [pYAP]$$

$R_5$ as phosphorylation:

$$R_3 = K_{phospho} [YAPn]$$

$R_6$ as dephosphorylation:

$$R_4 = K_{dephospho} [pYAPn]$$

The transport reactions for YAP and pYAP are defined as flux densities across the nuclear membrane that depends on the respective concentrations; these are translated into Neumann-type interface conditions for the inner boundary of the cytoplasm and the outer boundary of the nucleus by the simulator software.

1. Flux density of YAP import
   $$F_{import} = k_{import} [YAP]$$
2. Flux density of YAP export
   $$F_{export} = k_{export} [pYAPn]$$

Boundary conditions on the outer membrane of the cytoplasm are 'zero-flux' Neumann type for all variables.

## Model parameters

**Appendix 1—table 1.** Canonical model parameters.

| Parameter | Symbol | YAP model | TAZ model | Unit |
|---|---|---|---|---|
| Diffusion | $D_{YAP}$ $D_{YAPn}$ | 0.0391776 | 0.00812799 | $\mu m^2/s$ |
| Diffusion phosphorylated | $D_{pYAPn}$ $D_{pYAP}$ | 0.433785 | 0.501116 | $\mu m^2/s$ |
| Phosphorylation | $K_{phospho}$ | 3.2462731384467 | 9.2660885020866 | $s^{-1}$ |
| Dephosphorylation | $K_{dephospho}$ | 2.4380091829261 | 2.9721728333365 | $s^{-1}$ |
| Translation | $K_{translation}$ | 0.013295743091699 | 0.02329254213393 | a.u./s |
| Degradation | $K_{degradation}$ | 0.0081058910608829 | 0.0090941179957213 | $s^{-1}$ |
| Nuclear import | $K_{import}$ | 9.7229134833679e−16 | 8.1106772583928e−16 | $\mu m/s$ |
| Nuclear export | $K_{export}$ | 9.5119620861314e−17 | 6.1050891990604e−17 | $\mu m/s$ |

**Appendix 1—table 2.** Alternative model parameters.

| Parameter | Symbol | YAP model | TAZ model | Unit |
|---|---|---|---|---|
| Diffusion | $D_{YAP}$ $D_{YAPn}$ | 9.84707 | 9.14393 | $\mu m^2/s$ |
| Diffusion phosphorylated | $D_{pYAPn}$ $D_{pYAP}$ | 1.04518 | 1.00732 | $\mu m^2/s$ |
| Phosphorylation | $K_{phospho}$ | 0.028839815689838 | 0.10286291836829 | $s^{-1}$ |
| Dephosphorylation | $K_{dephospho}$ | 0.09121073484801 | 0.10517790271802 | $s^{-1}$ |
| Translation | $K_{translation}$ | 0.009403085138566 | 0.0030932046019853 | a.u./s |
| Degradation | $K_{degradation}$ | 0.0094299166245222 | 0.0097940321843269 | $s^{-1}$ |
| Nuclear import | $K_{import}$ | 6.5845623444598e−15 | 9.5850453479782e−15 | $\mu m/s$ |
| Nuclear export | $K_{export}$ | 7.8819897236132e−15 | 4.5876638224922e−15 | $\mu m/s$ |

**Appendix 1—table 3.** Parameters of the YAP-to-TAZ transition model.

| Parameter | Symbol | Value | Unit |
|---|---|---|---|
| Diffusion | $D_{YAP}$ $D_{YAPn}$ | 10 | $\mu m^2/s$ |
| Diffusion phosphorylated | $D_{pYAPn}$ $D_{pYAP}$ | 1 | $\mu m^2/s$ |
| Phosphorylation | $K_{phospho}$ | − | $s^{-1}$ |
| Dephosphorylation | $K_{dephospho}$ | | $s^{-1}$ |
| Translation | $K_{translation}$ | 0.01 | a.u./s |
| Degradation | $K_{degradation}$ | 0.004 | $s^{-1}$ |
| Nuclear import | $K_{import}$ | 7.5e−15 | $\mu m/s$ |
| Nuclear export | $K_{export}$ | 1.6e−15 | $\mu m/s$ |

**Appendix 1—table 4.** Parameter range for particle swarm algorithm of the canonical YAP/TAZ model.

| Parameter | Symbol | Range (min, max) | Unit |
|---|---|---|---|
| Diffusion | $D_{YAP}$ $D_{YAPn}$ | (0.001, 5) | µm²/s |
| Diffusion phosphorylated | $D_{pYAPn}$ $D_{pYAP}$ | (0.001, 1) | µm²/s |
| Phosphorylation | $K_{phospho}$ | (0.1, 10) | s⁻¹ |
| Dephosphorylation | $K_{dephospho}$ | (0.1, 10) | s⁻¹ |
| Translation | $K_{translation}$ | (1e−3, 0.1) | a.u./s |
| Degradation | $K_{degradation}$ | (1e−3, 1e−2) | s⁻¹ |
| Nuclear import | $K_{import}$ | (1e−16, 1e−15) | µm/s |
| Nuclear export | $K_{export}$ | (1e−17, 1e−15) | µm/s |

**Appendix 1—table 5.** Parameter range for particle swarm algorithm of the alternative YAP/TAZ model.

| Parameter | Symbol | Range (min, max) | Unit |
|---|---|---|---|
| Diffusion | $D_{YAP}$ $D_{YAPn}$ | (1, 10) | µm²/s |
| Diffusion phosphorylated | $D_{pYAPn}$ $D_{pYAP}$ | (1, 1.5) | µm²/s |
| Phosphorylation | $K_{phospho}$ | (0.01, 0.5) | s⁻¹ |
| Dephosphorylation | $K_{dephospho}$ | (0.01, 0.5) | s⁻¹ |
| Translation | $K_{translation}$ | (1e−3, 0.01) | a.u./s |
| Degradation | $K_{degradation}$ | (1e−3, 0.01) | s⁻¹ |
| Nuclear import | $K_{import}$ | (1e−16, 1e−14) | µm/s |
| Nuclear export | $K_{export}$ | (1e−16, 1e−14) | µm/s |

**Appendix 1—table 6.** Parameter values from the literature.

| Parameter | Value | Source |
|---|---|---|
| Hepatocyte cell volume | 10–11 l | *Jack et al., 1990* |
| Hepatocyte nucleus volume | $5 \times 10^{-13}$ | *Jack et al., 1990* |
| Nuclear export | 10–100 s | *Ege et al., 2018* |
| Nuclear import | 50 s | *Ege et al., 2018* |
| Diffusion rate YAP/TAZ | 19 µm²/s | *Ege et al., 2018* |

