## [Editor Report]

The Hippo signaling pathway is essential for multiple physiological processes, most notably the regulation of cell proliferation and survival during wound healing. Wehling et al. provide a molecular framework for an alternative mechanism by which the Hippo effector molecule YAP's sub-cellular localization is regulated by cell compartment-specific phosphorylation. Specifically, the authors demonstrate dynamic regulation of shuttling of YAP both in vitro and in vivo during drug-induced liver injury. Given the importance and developmental conserveness of the Hippo pathway, the work is of broad interest to the field of developmental and regenerative biology.

---

## [Decision Letter]

**Decision letter after peer review:**

Thank you for submitting your article "Spatial modeling reveals nuclear phosphorylation and subcellular shuttling of YAP upon drug-induced liver injury" for consideration by *eLife*. Your article has been reviewed by 3 peer reviewers, including Matthew A Quinn as the Reviewing Editor and Reviewer #1, and the evaluation has been overseen by a Reviewing Editor and Mone Zaidi as the Senior Editor. The following individual involved in the review of your submission has agreed to reveal their identity: Dirk Fey (Reviewer #2).

Essential revisions:

1) The model features very different diffusion rates for the canonical and alternative model, but is not explained why. For the estimation, different parameter ranges were specified for the canonical and alternative model, also for other parameters to be estimated, e.g. phosphorylation and dephosphorylation rates. This also lacks an explanation. What is the rationale behind this? Please update the discussion and/or methods to clarify this issue.

2) 200 parametrizations were obtained. Is the simulation result similar? Maybe include a panel in figure S2 in the supplement about this. Also, include a table with all fitted parameters.

3) Although of a dynamic nature, the model was not analyzed for the time-dependency of its simulations nor compared to time-course data, and the manuscript would benefit from a brief discussion, in particular with respect to the time-dependent biphasic response.

4) In Figures 2B and 2D, please highlight the boundary between nucleus and cytoplasm in the pictures, as the main difference between the two modeling results is the direction of gradient around the nucleus-cytoplasm boundary. In addition to the modeling result confined by the parameter space, please also provide an intuitive explanation of why the canonical model cannot account for the discrepancies between experiment and simulation.

5) Figure 3 on APAP regulation of YAP phosphorylation: As APAP generally induces toxicity, it makes sense to quantify cell death in the culture and limit the measurement to live cells. If possible, it will be nice to monitor the same cell population over the course of APAP or control treatment, and see how the nuclear/cytoplasmic ratio of YAP and TAZ change as a function of a cell's local environment. Also, did the proposed YAP phosphorylation in early APAP treatment (<6h) lead to YAP shuttling inside the nuclei?

6) The western immunoblot bands are hard to be interpreted by eye except when showing a binary result. Please add quantification for the plots in Figure 3C and Figure 4A, C, D, E, and F.

*Reviewer #1 (Recommendations for the authors):*

This is an interesting and thorough investigation of the regulation of Yap localization via its nuclear phosphorylation. While this is a novel concept and well-supported by the data provided, there are several unanswered questions that need to be resolved either by additional experiments or by an enhanced discussion as detailed below:

1) In figure 2, the authors demonstrate kinases able to phosphorylate Yap (LATS1/2) display a nuclear localization under low cell seeding conditions. However, is this localization of LATS1/2 dependent on cell density? Does a LATS1/2 knockdown or overexpression affect Yap phosphorylation and shuttling? Does APAP affect LATS1/2 localization? Answering these questions experimentally will give more credence to the claim that Yap regulatory machinery is located in the nucleus and affords functional consequences to those regulatory proteins.

2) Figure 4 shows the quantification of duolink particles for various proteins under control and APAP groups. However, the APAP treated group is not displayed. A representative image of the APAP duolink is needed.

3) From a mechanistic perspective the authors utilize an Akt inhibitor and show dephosphorylation of YAP. They also show that Akt inhibitors block APAP-induced phosphorylation. However, the authors do not determine the effects of the downstream sub-cellular localization of Yap. Determining if Akt inhibitors affect Yap localization during APAP treatment is needed to make a mechanistic link between APAP/Akt/Yap translocation.

4) The authors show APAP induces Yap translocation in vivo during DILI. While this demonstrates in vivo relevance to their in vitro findings, it does not convey any functional relevance in the modulation of the regenerative response. Does Akt inhibition affect hepatic Yap translocation and downstream survival or regeneration? Performing additional in vivo experiments modulating the Akt/ROS/Yap pathway is essential in order to draw conclusions on whether this is a viable therapeutic target in vivo during DILI.

*Reviewer #2 (Recommendations for the authors):*

1) The model features very different diffusion rates for the canonical and alternative model, but is not explained why. For the estimation, different parameter ranges were specified for the canonical and alternative model, also for other parameters to be estimated, e.g. phosphorylation and dephosphorylation rates. This also lacks an explanation. What is the rationale behind this?

2) 200 parametrizations were obtained. Is the simulation result similar? Maybe include a panel in figure S2 in the supplement about this. Also, include a table with all fitted parameters.

3) Although of a dynamic nature, the model was not analyzed for the time-dependency of its simulations nor compared to time-course data, and the manuscript would benefit from a brief discussion, in particular with respect to the time-dependent biphasic response.

*Reviewer #3 (Recommendations for the authors):*

I believe the conclusions are mostly supported by the result. My comments are mostly on data presentation and interpretation:

1. In Figures 2B and 2D, please highlight the boundary between nucleus and cytoplasm in the pictures, as the main difference between the two modeling results is the direction of gradient around the nucleus-cytoplasm boundary. In addition to the modeling result confined by the parameter space, please also provide an intuitive explanation of why the canonical model cannot account for the discrepancies between experiment and simulation.

2. Figure 3 on APAP regulation of YAP phosphorylation: As APAP generally induces toxicity, it makes sense to quantify cell death in the culture and limit the measurement to live cells. If possible, it will be nice to monitor the same cell population over the course of APAP or control treatment, and see how the nuclear/cytoplasmic ratio of YAP and TAZ change as a function of a cell's local environment. Also, did the proposed YAP phosphorylation in early APAP treatment (<6h) lead to YAP shuttling inside the nuclei?

3. The western immunoblot bands are hard to be interpreted by eye except when showing a binary result. Please add quantification for the plots in Figure 3C and Figure 4A, C, D, E, and F.

Finally, it would be nice if the authors could discuss (1) the role of nuclear phosphorylation in cell proliferation, tissue damage, and regeneration – i.e. whether the YAP/TAZ localization is the cause or consequence of cell density change, and (2) the biological implication of context-dependent effect of APAP in early and late treatment.

---

## [Author Response]

Essential revisions:1) The model features very different diffusion rates for the canonical and alternative model, but is not explained why. For the estimation, different parameter ranges were specified for the canonical and alternative model, also for other parameters to be estimated, e.g. phosphorylation and dephosphorylation rates. This also lacks an explanation. What is the rationale behind this? Please update the discussion and/or methods to clarify this issue.

This question is answered in detail below (see Reviewer 2, comment #1).

2) 200 parametrizations were obtained. Is the simulation result similar? Maybe include a panel in figure S2 in the supplement about this. Also, include a table with all fitted parameters.

This question is answered in detail below (see Reviewer 2, comment #2).

3) Although of a dynamic nature, the model was not analyzed for the time-dependency of its simulations nor compared to time-course data, and the manuscript would benefit from a brief discussion, in particular with respect to the time-dependent biphasic response.

This question is answered in detail below (see Reviewer 2, our reply to the general comment as well as comment #3).

4) In Figures 2B and 2D, please highlight the boundary between nucleus and cytoplasm in the pictures, as the main difference between the two modeling results is the direction of gradient around the nucleus-cytoplasm boundary. In addition to the modeling result confined by the parameter space, please also provide an intuitive explanation of why the canonical model cannot account for the discrepancies between experiment and simulation.

This question is answered in detail below (see Reviewer 3, comment #1).

5) Figure 3 on APAP regulation of YAP phosphorylation: As APAP generally induces toxicity, it makes sense to quantify cell death in the culture and limit the measurement to live cells. If possible, it will be nice to monitor the same cell population over the course of APAP or control treatment, and see how the nuclear/cytoplasmic ratio of YAP and TAZ change as a function of a cell's local environment. Also, did the proposed YAP phosphorylation in early APAP treatment (<6h) lead to YAP shuttling inside the nuclei?

This question is answered in detail below (see Reviewer 3, comment #2).

6) The western immunoblot bands are hard to be interpreted by eye except when showing a binary result. Please add quantification for the plots in Figure 3C and Figure 4A, C, D, E, and F.

This question is answered in detail below (see Reviewer 3, comment #3).

Reviewer #1 (Recommendations for the authors):This is an interesting and thorough investigation of the regulation of Yap localization via its nuclear phosphorylation. While this is a novel concept and well-supported by the data provided, there are several unanswered questions that need to be resolved either by additional experiments or by an enhanced discussion as detailed below:1) In figure 2, the authors demonstrate kinases able to phosphorylate Yap (LATS1/2) display a nuclear localization under low cell seeding conditions. However, is this localization of LATS1/2 dependent on cell density? Does a LATS1/2 knockdown or overexpression affect Yap phosphorylation and shuttling? Does APAP affect LATS1/2 localization? Answering these questions experimentally will give more credence to the claim that Yap regulatory machinery is located in the nucleus and affords functional consequences to those regulatory proteins.

We thank reviewer for the suggestions. The mechanistic connection between LATS1/2 kinases and the Hippo pathway effectors has been intensively investigated in different species and cell types. To further emphasize the relevance of the LATS1/2-YAP connection in nuclei several experiments were initiated.

(A) To answer the question whether LATS1/2 localization is dependent on cell density, we performed nuclear-cytoplasmic fractionation experiments for low and high cell density conditions (Author response image 1). Indeed, cell density conditions affected LATS1 and LATS2 localization differently. The amount of nuclear LATS1 protein was increasing at high cell density (NCR 1.4 at low cell density to 4.1 at high cell density). In contrast, LATS2, which was predominantly localized in the cytoplasm, did not shuttle in a comparable manner. Phosphorylated LATS1/2 (pLATS1/2) increased in the nucleus at high cell density conditions (NCR from 0.7 to 1.1), indicating that the LATS-dependent regulatory processes take place at high cell density in cell nuclei. To sum up, these experiments demonstrate that nuclear LATS abundancy is indeed regulated by cell density.

(B) To answer the question whether LATS knockdown affects YAP phosphorylation and shuttling, we inhibited LATS1/2 expression using gene-specific siRNAs targeting LATS1 and LATS2 (Author response image 1). As illustrated by Western blot quantification, LATS2 silencing reduced phosphorylation of YAP about 30%, whereas LATS1 knockdown did not significantly alter pYAP abundance. Thus, both LATS proteins contribute YAP in our experimental setup differentially.

(C) We further quantitatively explored the localization of YAP after siLATS treatment using live-cell imaging (Author response image 1). Data illustrated significantly increased NCR for YAP in siLATS treated cells versus controls cells at low and high cell density conditions (see quantification).

Together with the data we showed in the first version of the manuscript (nuclear localization of LATS1/2 and physical interaction between active LATS1/2 and YAP in nuclei), we conclude that nuclear LATS is indeed an important player in regulating Hippo pathway.

**Author response image 1. sa2fig1:** Western blot of nuclear and cytoplasmic protein fractions derived from Hep3B cells under low and high cell density conditions (5x10^5^ for low cell density and 3x10^6^ cells for high cell density per 10 cm dish). Quantification of the NCR for LATS1, LATS2, and phosphorylated LATS (pLATS1/2) under low and high cell density is depicted on the right. C: cytoplasmic fraction; N: nuclear fraction. (B): Inhibition of LATS proteins in Hep3B cells using siRNAs at different concentrations for 4 hr (20 and 40 nM; (siLATS1, 5’-CCA GAA GGA UAU AGA CAA AdTdT-3’; siLATS2, 5’-CAU GAA GAC CCU AAG GAA AdTdT-3’). Quantification of LATS knockdown efficiency and effect on YAP phosphorylation is shown on the right. Ctrl: untreated control, NTC: no template control. Cytoplasmic (C) and nuclear (N) protein fraction. (C): Live-cell imaging of Hep3B cells after LATS inhibition (siLATS1/2) and NTC (no template control) treatment (left). Scale bar 50 µm. Quantification of the NCR between NTC (n=60) and siLATS (n=55) with respect to cell count per image (representing cell density). Shaded area around the green and yellow curves represent 96% confidence interval of the fitted log(x) function. Statistics: two-tailed Mann-Whitney test with p-value<0.0001.

(D) Finally, we aimed to answer the question whether APAP affects LATS localization in our model system (Author response image 2). In cytoplasmic-nuclear fractionation experiments we observed that APAP treatment causes a moderate decrease in nuclear fraction of LATS1. Here, LATS2 localization was strongly influenced by APAP treatment, especially after 48 hours of treatment (NCR reduced about 60% in comparison to control). In addition, the localization of the phosphorylated LATS isoform (pLATS1/2) was also regulated by APAP. In general, APAP treatment led to reduced pLATS in nuclei and cytoplasm. Taken together, this data illustrates that APAP treatment reduced total amount of LATS1, LATS2 and pLATS1/2 proteins and also induced the cytoplasmic shift for those proteins. Thus, LATS proteins are less active after APAP administration and may contribute to the observed less efficient YAP phosphorylation.

**Author response image 2. sa2fig2:** Western blotting of cytoplasmic (C) and nuclear (N) fraction 2 and 48 hours after APAP (10 mM) and control (ctrl) treatment in Hep3B cells. The localization of LATS1, LATS2 and pLATS1/2 proteins was detected. PARP and Tubulin served as loading controls for nucleus and cytoplasm, respectively. The quantification of the NCR of LATS1, LATS2 and pLATS1/2 shown on the right. Western blots were obtained from one experiment.

In summary, the performed experiments confirming that nuclear LATS1/2 protein are of importance for YAP shuttling. In addition, APAP (at least partly) may contribute to the subcellular localization of YAP via the regulation/exclusion of LATS1/2 from hepatocellular cell nuclei.

Because this information is of relevance for the readership, a clarifying statement was added in the revised version of the manuscript: **"** Indeed, we and others showed that nuclear LATS1/2 can control YAP phosphorylation and localization (data not shown) (Ege et al., 2018; Li et al., 2014). The relevance of this nuclear interaction is substantiated by a proximity ligation assay (PLA), which illustrated that pLATS1/2 physically interacted with YAP not only in the cytoplasm but also in the nucleus (Figure 2G)." Page 7, lines 251-256

2) Figure 4 shows the quantification of duolink particles for various proteins under control and APAP groups. However, the APAP treated group is not displayed. A representative image of the APAP duolink is needed.

We thank the reviewer for the suggestion. According to the comment, we now added representative images for the experiment in Figure 4I to the manuscript.

3) From a mechanistic perspective the authors utilize an Akt inhibitor and show dephosphorylation of YAP. They also show that Akt inhibitors block APAP-induced phosphorylation. However, the authors do not determine the effects of the downstream sub-cellular localization of Yap. Determining if Akt inhibitors affect Yap localization during APAP treatment is needed to make a mechanistic link between APAP/Akt/Yap translocation.

We thank the reviewer for the suggestion. Indeed, the mechanistic link between AKT and YAP was not intensively investigated in the first manuscript draft.

Before we present/discuss our new results, we want to emphasize that long-term experiments with AKTi (24 or 48 hours) were difficult to perform due to (unspecific?) cell toxic effects of the inhibitor. Instead, all experiments were performed at early time points (up to 3 hours), which is of special importance when interpreting those experiments where APAP is equally given (early APAP is leading to YAP phosphorylation, see figure 3C of main manuscript).

Since the reviewer asked for the effects of AKTi on YAP localization, we also performed short- and long-term life cell imaging experiments using fluorescently-tagged YAP after APAP and AKTi treatment. However, it turned out that the results couldn’t be analyzed for technical reasons. In detail, combined APAP/AKTi treatment disintegrated the signal of the nuclear marker (H2B-tagged cerulean), which prevented the quantitative analysis of nuclear and cytoplasmic YAP (Author response image 3). Indeed, the AKT-dependent regulation of DNA fragmentation and histone deacetylase proteins has been described before (Ahn et al., 2006; Ozaki et al., 2010). Because the impaired nuclear signal quality led to unreliable interpretation of the live-cell imaging results, these results cannot be used to substantiate our findings.

Based on the data derived from experiments shown in Author response image 3/B we conclude that AKTi leads to a (moderate) nuclear localization of YAP early (!) after APAP treatment. Thus, blocking AKT at early time points after APAP treatment prevented YAP phosphorylation (which is observed at early time-points) and therefore reduced nuclear exclusion.

We adjusted a statement in the revised manuscript: "Indeed, blocking AKT activity by AKTi reduced YAP phosphorylation, moderately elevated nuclear YAP positivity, and increased the physical interaction between AKT and YAP in cell nuclei (Figure 4F and data not shown)." Page 9, line 351 – page 10, line 354

**Author response image 3. sa2fig3:** (A): Left: nuclear and cytoplasmic protein fractionation after APAP (10 mM, 1h in FCS-free medium) and combined APAP and AKTi (AKT inhibitor VIII, 10 µM 3h in FCS-free medium) treatment of Hep3B cells. PARP and Tubulin serve as loading controls for nuclear and cytoplasmic fraction, respectively. Right: quantification of the NCR of YAP after APAP and combined APAP and AKTi treatment. (B): DuoLink PLA between YAP and AKT proteins under APAP (10 mM, 2h) and combined APAP and AKTi (10 µM, 3h) treatment. Top row: nuclei (blue) and YAP-AKT interaction (red dots); bottom row: segmentation of the nuclei and PLA dots. Right: quantification of the PLA dots in nuclei for APAP (27 images) and combined APAP/AKTi treatment (31 images). Statistics: two tailed Mann-Whitney test, p-value<0.0001. Scale bar 50 µm. (C): Live-cell imaging of Hep3B cells under APAP (10 mM 2h, top row) and combined APAP and AKTi (10µM 3h, bottom row) treatment.

4) The authors show APAP induces Yap translocation in vivo during DILI. While this demonstrates in vivo relevance to their in vitro findings, it does not convey any functional relevance in the modulation of the regenerative response. Does Akt inhibition affect hepatic Yap translocation and downstream survival or regeneration? Performing additional in vivo experiments modulating the Akt/ROS/Yap pathway is essential in order to draw conclusions on whether this is a viable therapeutic target in vivo during DILI.

We thank the reviewer for the suggestion to further investigate the therapeutic relevance of AKT and YAP inhibition during APAP-induced DILI. We discussed this aspect with our collaboration partners (A. Ghallab and J. Hengstler, Department of Toxicology, Dortmund). However, we decided not to perform e.g., AKT inhibition experiments in vivo for ethical and experimental reasons:

First, the inhibitor used in our study is very potent and efficiently abrogates AKT_1/2_ activity (see main Figure 4E). Although the genetic inactivation of AKT_1/2_ in specific cell types such as cardiomyocytes can be investigated in vivo, the general blockade of AKT_1/2_ can cause rapid lethality as illustrated in newborn and adult mice with AKT1- and AKT2-deficiency (Liu et al., 2019; Peng et al., 2003; Wang et al., 2016). Thus, it is likely that the complete inhibition of both AKT isoforms by using an efficient chemical inhibitor may cause unpredictable side effects or is not compatible with life.Second, as seen for APAP administration, the genetic inhibition of AKT isoforms in liver tissue cause severe hepatotoxicity and liver injury (Black, 1984; Wang *et al.*, 2016). This strongly suggests that the treatment with APAP and concomitant silencing of AKT isoforms would further intensify hepatotoxicity leading to the unpredictable behavior of YAP expression and localization.

Due to these obstacles, we believe that these experiments are ethically problematic and very likely do not provide generate meaningful results.

However, we decided to perform additional mouse experiments to substantiate our experimental conclusions that AKT supports YAP activity in hepatocytes. For this, we did hydrodynamic gene delivery experiments in vivo with vectors coding for constitutively active myristoylated AKT (myrAKT) (Luiken et al., 2020). According to our hypothesis, permanent myrAKT activity should support nuclear YAP enrichment followed by the induction of typical transcriptional target genes (e.g., Ctgf, Cyr61). Indeed, immunohistochemical (IHC) tissue staining of extracted mice liver illustrated not only illustrated elevated AKT expression followed by hepatocellular steatosis (Wang et al., 2015) but also nuclear YAP positivity. Moreover, both investigated target genes were induced at the transcript level in mice after myrAKT injection (Author response image 4/B).

**Author response image 4. sa2fig4:** (A): Immuno-histochemical staining of AKT (left) and YAP (right) of tissues derived from FVB mice after hydrodynamic injection of expression vectors coding myrAKT. Experiments and ethical aspects have been published previously (Luiken *et al.*, 2020). Scale bar 50 µm. (B): Quantitative polymerase chain reaction (qPCR) measurements of murine Hippo target genes Ctgf and Cyr61 (biological replicates: n_control_ = 3, n_myrAKT_ = 4). Statistics: unpaired two tailed t test; p-value_Ctgf_ = 0.049, p-value_Cyr61_ = 0.026.

The results confirm that AKT support nuclear YAP enrichment. To keep the manuscript concise, we suggest to include the following clarifying statement in the manuscript: "The mechanistic connection between AKT activity and YAP induction in murine hepatocytes was confirmed in independent experiments. Here hydrodynamic gene delivery of myristoylated AKT led to nuclear enrichment of YAP expression in hepatocytes and expression of typical target genes (data not shown)." in paragraph ' Sequential activation of ROS, AKT, and Hippo/YAP in mouse livers after APAP intoxication'. Page 11, lines 402-407

**Author response table 1. sa2table1:** Murine primers used for qPCR. Expression of Cyr61 and Ctgf were normalized to a set of house-keeping genes: Actin, Gapdh, Tbp, Tubulin, Hprt, Ppia (software: GeNorm).

Gene	Forward	Reverse
Actin	GCTTCTTTGCAGCTCCTTCGT	ACCAGCGCAGCGATATCG
Gapdh	TGTCCGTCGTGGATCTGAC	CCTGCTTCACCACCTTCTTG
Tbp	TTGTCTGCCATGTTCTCCTG	CAGGGTGATTTCAGTGCAGA
Tubulin	TCACTGTGCCTGAACTTACC	GGAACATAGCCGTAAACTGC
Ctgf	GGAGAACTGTGTACGGAGCG	CCAGGCAAGTGCATTGGTA
Hprt	TCCTCCTCAGACCGCTTTT	CCTGGTTCATCATCGCTAATC
Ppia	GCATACAGGTCCTGGCATCT	AGCTGTCCACAGTCGGAAAT
Cyr61	GATCTGTGAAGTGCGTCCTTGTGG	GACACTGGAGCATCCTGCATAAG

Reviewer #2 (Recommendations for the authors):1) The model features very different diffusion rates for the canonical and alternative model, but is not explained why. For the estimation, different parameter ranges were specified for the canonical and alternative model, also for other parameters to be estimated, e.g. phosphorylation and dephosphorylation rates. This also lacks an explanation. What is the rationale behind this?

This comment/question involves two aspects: 1) Why are parameters that should be the same physical constant in both models different (e.g. diffusion coefficients)? 2) Why did we choose different value ranges for the fitting procedure in the different models?

In this work it was not our intention to exactly determine the parameter values. Indeed, all parameters are known to be structurally unidentifyable since only stationary data with arbitrary concentration units is used for the fitting. Choosing a different time scale for the model would change all parameter values while still resulting in the same steady state concentration pattern. This means the actual parameter values are *de facto* arbitrary. Instead, we focus on the qualitative result of our modeling: one version of the model cannot fit the data (given a wide range of allowed parameter values) while the other version can. Identical diffusion constants in both model versions would not strengthen our conclusion. Specifically, we have shown that the simple model without the same diffusion constants cannot reproduce the observations. Thus, adding an additional restriction on the diffusion constants will definitely not lead to a better fitting model.

The reason for different parameter ranges in both models is the numeric stability of the PDE solver for the canonical model. Extreme numeric values for the diffusion constants lead to very stiff models, which complicate PDE simulations. Therefore, parameter values for the canonical model were chosen so that the complete range could be reliably simulated for parameter fitting. As explained above, the actual parameter values are not identifyable and not the scope of this study. It is important that the range is big enough that a good fit for the simple model was not arbitrarily excluded. From our experience we are convinced that this condition is satisfied. We agree that identical ranges for both models would be 'nicer' and future versions of our simulator software may offer this possibility. However, for the reasons mentioned above we are convinced that our results are sound and can explain the biological observations sufficiently.

We want to emphasize that even in the absence of non-identifyability the parameter values resulting from a 'failed' fitting procedure are not physically meaningful. We have shown with extensive parameter estimation runs that we were not able to generate at least one single satisfying fit for the canonical model. Any specific parameter values from these unsuccessful fitting runs, even if not affected by structural non-identifyability, are inherently meaningless.

In order to clarify the discrepancies between parameters of canonical and alternative models, we added the following paragraph to our methods sections:

“The parameter ranges of our canonical and alternative models were selected in a way to describe biologically meaningful value ranges and to avoid numeric instability during PDE simulations of the canonical model.” Page 25, lines 842-844

2) 200 parametrizations were obtained. Is the simulation result similar? Maybe include a panel in figure S2 in the supplement about this. Also, include a table with all fitted parameters.

According to the reviewer's comment, we now provide a csv file with all obtained parameter values for each alternative model and the objective functions of the models. See Supplementary File 1 (for YAP) and 2 (for TAZ).

In addition, we provide additional information on distribution of the objective function across models (Figure 2—figure supplement 1A) and parameter values of the 30 best models (Figure 2—figure supplement 1B). Here, we overlaid the values of the 30 best models (based on the objective function minimum value after 200 iterations) with the parameter values form the rest of the models. In general, this illustrates the structural non-identifyability of all parameters in the models (as discussed in the manuscript), which is not a problem since we focus on the ability of the various model variants to fit the data rather than the values of the specific parameters. Interestingly, some specific parameters, such as, phosphorylation reaction in the nucleus (both models YAP and TAZ) and degradation parameter for TAZ models, seem to cluster in the 'good fit', possibly indicating the importance of these parameter values for the fit. Lastly, we show exemplarily how a “bad fit” looks in comparison to a 'good fit' (Author response image 5)

**Author response image 5. sa2fig5:** A visual representation of one of the best models for YAP and TAZ, and the worst models. Parameter value distribution of YAP.

3) Although of a dynamic nature, the model was not analyzed for the time-dependency of its simulations nor compared to time-course data, and the manuscript would benefit from a brief discussion, in particular with respect to the time-dependent biphasic response.

We thank the reviewer for pointing out the time-dependency as an important player in Hippo signaling dynamics. This suggestion could indeed initiate an interesting and challenging follow up study. Model fitting and parameter estimation for spatial models is challenging even for not time-dependent data (indeed not many examples exist in the published literature) and using time-dependent data for fitting is not yet implemented in our software platform.

The reviewer is correct when he/she emphasizes that simulation of the time dependency could answer complex questions of dynamic nature. However, the initial question of our study (which pathway topology explains the data) was tailored to steady state models. A more complex model with further assumptions would indeed expand the view on the Hippo pathway, however, this would require further intensive work which would not significantly affect the outcome of our study and therefore could not be in the scope of the manuscript.

To clarify our aims regarding steady state PDE spatial modelling, we added the following passage to the discussion paragraph:

“The goal of our approach was to determine whether nuclear phosphorylation of YAP plays a role in Hippo signaling, specifically in governing YAP localization. A model not including nuclear phosphorylation of YAP could not sufficiently explain the experimentally observed localization pattern and was excluded from further analyses. However, the alternative model proposed in this work, can explain the experimental data and therefore establishes a possible mechanism for YAP localization. However, more complex mechanisms regulating subcellular localization and dynamics of YAP and TAZ cannot be excluded. Therefore, our model partly explains how the Hippo pathway is organized, however, it does not aim to comprehensively describe it." Page 12, lines 430-440

Reviewer #3 (Recommendations for the authors):I believe the conclusions are mostly supported by the result. My comments are mostly on data presentation and interpretation:1. In Figures 2B and 2D, please highlight the boundary between nucleus and cytoplasm in the pictures, as the main difference between the two modeling results is the direction of gradient around the nucleus-cytoplasm boundary. In addition to the modeling result confined by the parameter space, please also provide an intuitive explanation of why the canonical model cannot account for the discrepancies between experiment and simulation.

We thank the reviewer for the suggesting and now added a thin contour line around the nuclear border to highlight boundaries between cytoplasm and nuclei in main Figure 2B and 2D.

In addition, we agree with the reviewer that the inappropriateness of the canonical model (and why the extended model is more appropriate) can be explained better. In detail, the measured distribution of YAP and TAZ would imply diffusion out of the nucleus due to a gradient formed from nucleus to cytosol. Since YAP/TAZ are not synthesized in the nucleus, a second chemical species with different diffusion coefficients has to be exported out of the nucleus (the phosphorylated version of the protein). This prerequisite cannot be fulfilled by the canonical model.

This is obvious considering the residuals of the canonical model that clearly show discrepancy between simulation and the experimental results. Especially in areas close to the nuclear envelope in figure 2B/D, the canonical model is underrepresenting YAP/TAZ concentrations as depicted by deep blue pixels around the nucleus. Furthermore, the canonical model does not correctly represent the nuclear distribution of YAP and TAZ. Here, nuclear YAP and TAZ are underrepresented in the nuclear center (blue pixels), which is an inherent property of the canonical model topology.

To clarify the process behind the model selection, we added/adjusted the following passages in the results chapter of our manuscript:

“For example, the model cannot sufficiently explain the distribution of YAP and TAZ in cell nuclei or at the nuclear membrane. In detail, the cell area around the nuclear envelope on the cytoplasmic side strongly underrepresented the concentration of YAP/TAZ, as indicated by the blue pixels in the residual image. Moreover, the simulated YAP/TAZ distribution pattern of the canonical model showed a decreased protein concentration in the center of the nucleus, with increasing gradient towards the nuclear envelope. This indicates that the observation is dominated by a protein that is disseminated from the nucleus and undergoes diffusion and degradation. The canonical Hippo pathway cannot explain this effect, illustrated by residuals (experimental data minus model simulation) (Figure 2B)." Page 6, lines 203-214

2. Figure 3 on APAP regulation of YAP phosphorylation: As APAP generally induces toxicity, it makes sense to quantify cell death in the culture and limit the measurement to live cells. If possible, it will be nice to monitor the same cell population over the course of APAP or control treatment, and see how the nuclear/cytoplasmic ratio of YAP and TAZ change as a function of a cell's local environment. Also, did the proposed YAP phosphorylation in early APAP treatment (<6h) lead to YAP shuttling inside the nuclei?

Topic: Toxicity

The reviewer is right when he/she underlines the cell-toxic properties of APAP, which may differ in cell culture models. To investigate this aspect in more detail, we performed three different experimentals: CellTox assay, Western blotting of cleaved PARP fragment, and live-cell imaging.

We measured time-dependent cell-toxicity after APAP (10 mM) treatment with a CellTox Green Cytotoxicity Assay (Promega, # G8742, Author response image 6). Here, APAP treatment for 48 hours moderately increased cell toxicity about 10% on average, whereas at 24 hours post treatment slightly decreased toxicity was observed. Although, mild toxicity is detectable in our experimental setup, we can conclude that the majority of cells is still viable at all time points of our experiments.We quantified cleaved cytoplasmic PARP 89 kDa fragments whose presence indicate cell death (Author response image 6). This experiment confirm that the amount of cleaved PARP slightly increased after APAP treatment for 48 hours.More important, for all live-cell imaging experiments, which are the basis for our study and mathematical modeling, only viable (attached) cell populations with clear nuclear/cytoplasmic structure were used (see main Figures 3 and Figure 3—figure supplement 1A/B). Thus, quantification of the nuclear to cytoplasmic ratio was exclusively performed on viable cells, since rounded and floating (dead) cells were excluded from the classification algorithm.

These results illustrate a weak and neglectable role of APAP on cell toxicity and cell death in our experimental setup. Because the chosen live cell approach excludes damaged and/or apoptotic cells, we conclude that our experimental setup allows an unbiased view on APAP-induced effects on cell signaling.

An explanatory statement summarizing the results is now included in the revised manuscript: "Severe effects of APAP on cell toxicity and apoptosis in the chosen experimental setup were excluded by measuring cell viability and PARP cleavage (data not shown)." in paragraph 'APAP regulates YAP protein localization and activity'. Page 8, lines 281-283

Topic: early response

As to early APAP treatment (<6 hours), we indeed did not explicitly mention YAP localization in our previous manuscript, since we believe that 'later' time points better describe the biologically relevant APAP responses of a cell.

To answer the reviewer's question, we approached this aspect with two different experimental techniques: live-cell imaging and Western immunoblotting (Author response image 6 and 6C, respectively).

()Live-cell imaging illustrates no significant nuclear/cytoplasmic shift of YAP after APAP treatment for 2 hours (Author response image 6). In this experimental setup, we also investigated different cell density conditions to exclude this aspect as important parameter.(B)These results indicate no or very weak cytoplasmic translocation of YAP. A possible explanation for this mild effect after short-term treatment of APAP might be the existence of molecular mechanisms (e.g., phosphorylation of alternative serine residues of YAP), which are not part of the current model. Due to the weak character of the results and the fact that we cannot offer a reasonable explanation for this observation, we would appreciate not to include these results in the revised version of the manuscript.

However, this aspect was already mentioned in the first submitted version of the manuscript. We wrote: " No obvious response was detectable for YAP and TAZ at earlier time points post APAP treatment (data not shown)." Page 8, lines 298-281

**Author response image 6. sa2fig6:** (A): CellTox assay quantifying non-viable cells in Hep3B cells after APAP (10 mM) treatment for 24 and 48 hours (n=10, technical replicates, independent biological replicates are not shown). Statistics: two-tailed unpaired t test, p-value<0.0001. (B): Nuclear/cytoplasmic protein fractioning of Hep3B cells after APAP (10mM) treatment for 2 and 48 hours. Total (116 kDa) and cleaved PARP (89 kDa) were detected. Tubulin served as fractionation control. Quantification of cleaved PARP fragments is shown below. C: cytoplasmic; N: nuclear fraction. (C): Live-cell imaging of Hep3B cells expressing Venus-tagged YAP 2 hours after APAP (10 mM) or control treatment (PBS). Exemplary fluorescent images (left) and data quantification of NCR as a function of cell count per image (right) are shown. One (out of three) representative experiment is shown (n_ctrl_ = 68; n_APAP_ = 68). Scale bar 50 µm. (D): Nuclear/cytoplasmic protein fractionation after APAP treatment (10 mM, 2 hours). Detection of YAP and pYAP is shown. PARP and tubulin serve as loading controls. Quantification of NCR between YAP and pYAP is shown (right). C: cytoplasmic; N: nuclear fraction. Western immunoblots in (B) and (D) were obtained from one experiment.

3. The western immunoblot bands are hard to be interpreted by eye except when showing a binary result. Please add quantification for the plots in Figure 3C and Figure 4A, C, D, E, and F.

According to the reviewer's suggestion, we now included quantification of all blots of the revised version of the manuscript (Figure 3—figure supplement 2).

Finally, it would be nice if the authors could discuss (1) the role of nuclear phosphorylation in cell proliferation, tissue damage, and regeneration – i.e. whether the YAP/TAZ localization is the cause or consequence of cell density change, and (2) the biological implication of context-dependent effect of APAP in early and late treatment.

According to the reviewer's suggestion, two additional paragraphs were included in the Discussion chapter of the revised manuscript. The first paragraph focuses on the aspect how nuclear YAP phosphorylation can extent our current thinking about Hippo pathway biology. Whereas the second paragraph discusses our findings in the context of biological/medical implications after long-term APAP ingestion. For both paragraphs additional references were included in the manuscript.

“The extension of the canonical Hippo pathway model with processes that control active nuclear YAP phosphorylation illustrates that the regulatory YAP/TAZ phosphorylation must be considered as continuum process (Shreberk-Shaked and Oren, 2019). For instance, the existence of nuclear phosphorylation possibilities by LATS-dependent and independent (e.g., by AKT) mechanisms increases the complexity but also flexibility and redundancy of the signaling pathway (Ege et al. 2018, Gao et al. 2017, Low et al. 2017). These cellular aspects are part of a fine-tuned regulatory network that allows a rapid and adjustable proliferative response of YAP and TAZ under diverse physiological and pathological cellular conditions (e.g., upon induction of tissue regeneration).” Page 13, lines 483-493

“In our manuscript we show that APAP administration induces a chain of molecular events through the APAP/ROS/AKT axis, which is leading to YAP inactivation (early) followed by YAP activation (late). The late induction of YAP activity might be considered as cell protective cellular response upon long-term tissue damage; however, YAP also acts as a potent oncogene in different cell types, including hepatocytes, and its overexpression is associated with tumor initiation (Dong et al. 2007; Weiler et al. 2017). Thus, it is tempting to speculate that APAP overdose may not only cause DILI but also increases the risk for liver tumor development. Although controversially discussed in the literature, a recent evaluation of 139 published epidemiologic studies strongly argues against a relationship between APAP exposure and cancer (Weinstein et al. 2021). Actually, APAP has been discussed as therapeutic option for patients with YAP-induced cancer (Poudel et al. 2021). Although, our and other studies clearly demonstrate a mechanistic link between APAP update and Hippo/YAP pathway activity, these partly controversial findings illustrate the necessity to decipher this connection under distinct disease conditions in the future." Page 15, lines 538-554

References

Ahn JY, Liu X, Liu Z, Pereira L, Cheng D, Peng J, Wade PA, Hamburger AWYe K. 2006. Nuclear Akt associates with PKC-phosphorylated Ebp1, preventing DNA fragmentation by inhibition of caspase-activated DNase. *EMBO J* 25:2083-2095. doi:10.1038/sj.emboj.7601111.

Black M. 1984. Acetaminophen hepatotoxicity. *Annu Rev Med* 35:577-593. doi:10.1146/annurev.me.35.020184.003045.

Ege N, Dowbaj AM, Jiang M, Howell M, Hooper S, Foster C, Jenkins RPSahai E. 2018. Quantitative Analysis Reveals that Actin and Src-Family Kinases Regulate Nuclear YAP1 and Its Export. *Cell Syst* 6:692-708 e613. doi:10.1016/j.cels.2018.05.006.

Li W, Cooper J, Zhou L, Yang C, Erdjument-Bromage H, Zagzag D, Snuderl M, Ladanyi M, Hanemann CO, Zhou P, Karajannis MAGiancotti FG. 2014. Merlin/NF2 loss-driven tumorigenesis linked to CRL4(DCAF1)-mediated inhibition of the hippo pathway kinases Lats1 and 2 in the nucleus. *Cancer Cell* 26:48-60. doi:10.1016/j.ccr.2014.05.001.

Liu W, Jing ZT, Xue CR, Wu SX, Chen WN, Lin XJLin X. 2019. PI3K/AKT inhibitors aggravate death receptor-mediated hepatocyte apoptosis and liver injury. *Toxicol Appl Pharmacol* 381:114729. doi:10.1016/j.taap.2019.114729.

Luiken S, Fraas A, Bieg M, Sugiyanto R, Goeppert B, Singer S, Ploeger C, Warsow G, Marquardt JU, Sticht C, De La Torre C, Pusch S, Mehrabi A, Gretz N, Schlesner M, Eils R, Schirmacher P, Longerich TRoessler S. 2020. NOTCH target gene HES5 mediates oncogenic and tumor suppressive functions in hepatocarcinogenesis. *Oncogene* 39:3128-3144. doi:10.1038/s41388-020-1198-3.

Ozaki K, Kosugi M, Baba N, Fujio K, Sakamoto T, Kimura S, Tanimura SKohno M. 2010. Blockade of the ERK or PI3K-Akt signaling pathway enhances the cytotoxicity of histone deacetylase inhibitors in tumor cells resistant to gefitinib or imatinib. *Biochem Biophys Res Commun* 391:1610-1615. doi:10.1016/j.bbrc.2009.12.086.

Peng XD, Xu PZ, Chen ML, Hahn-Windgassen A, Skeen J, Jacobs J, Sundararajan D, Chen WS, Crawford SE, Coleman KGHay N. 2003. Dwarfism, impaired skin development, skeletal muscle atrophy, delayed bone development, and impeded adipogenesis in mice lacking Akt1 and Akt2. *Genes Dev* 17:1352-1365. doi:10.1101/gad.1089403.

Shreberk-Shaked MOren M. 2019. New insights into YAP/TAZ nucleo-cytoplasmic shuttling: new cancer therapeutic opportunities? *Mol Oncol* 13:1335-1341. doi:10.1002/1878-0261.12498.

Wang Q, Yu WN, Chen X, Peng XD, Jeon SM, Birnbaum MJ, Guzman GHay N. 2016. Spontaneous Hepatocellular Carcinoma after the Combined Deletion of Akt Isoforms. *Cancer Cell* 29:523-535. doi:10.1016/j.ccell.2016.02.008.

Wang YH, Viscarra J, Kim SJSul HS. 2015. Transcriptional regulation of hepatic lipogenesis. *Nat Rev Mol Cell Bio* 16:678-689. doi:10.1038/nrm4074